# Erythrocyte invasion-neutralising antibodies prevent *Plasmodium falciparum* RH5 from binding to basigin-containing membrane protein complexes

Abhishek Jamwal[1,2], Cristina F Constantin[3], Stephan Hirschi[1,2], Sebastian Henrich[3†], Wolfgang Bildl[3], Bernd Fakler[3,4], Simon J Draper[1,2], Uwe Schulte[3,4], Matthew K Higgins[1,2]*

[1]Department of Biochemistry, University of Oxford, Oxford, United Kingdom; [2]Kavli Institute for Nanoscience Discovery, Dorothy Crowfoot Hodgkin Building, University of Oxford, Oxford, United Kingdom; [3]Institute of Physiology, Faculty of Medicine, University of Freiburg, Freiburg, Germany; [4]Signalling Research Centres BIOSS and CIBS, Freiburg, Germany

*For correspondence:
matthew.higgins@bioch.ox.ac.uk

Present address: †Roche Pharma AG, Emil-Barell Strasse 1, Grenzach-Whylen, Germany

**Abstract** Basigin is an essential host receptor for invasion of *Plasmodium falciparum* into human erythrocytes, interacting with parasite surface protein PfRH5. PfRH5 is a leading blood-stage malaria vaccine candidate and a target of growth-inhibitory antibodies. Here, we show that erythrocyte basigin is exclusively found in one of two macromolecular complexes, bound either to plasma membrane $Ca^{2+}$-ATPase 1/4 (PMCA1/4) or to monocarboxylate transporter 1 (MCT1). PfRH5 binds to each of these complexes with a higher affinity than to isolated basigin ectodomain, making it likely that these are the physiological targets of PfRH5. PMCA-mediated $Ca^{2+}$ export is not affected by PfRH5, making it unlikely that this is the mechanism underlying changes in calcium flux at the interface between an erythrocyte and the invading parasite. However, our studies rationalise the function of the most effective growth-inhibitory antibodies targeting PfRH5. While these antibodies do not reduce the binding of PfRH5 to monomeric basigin, they do reduce its binding to basigin-PMCA and basigin-MCT complexes. This indicates that the most effective PfRH5-targeting antibodies inhibit growth by sterically blocking the essential interaction of PfRH5 with basigin in its physiological context.

## Editor's evaluation

This elegantly performed and rigorous study generates new and conceptually important insights into the interaction between an essential malaria parasite invasion ligand (and vaccine candidate) called PfRH5, and its erythrocyte surface integral membrane receptor basigin. The authors provide compelling evidence based on rigorous biochemical assays that erythrocyte basigin is predominantly expressed in a complex with one of two distinct erythrocyte membrane proteins called PMCA and MCT1 and that PfRH5 binds to these complexes better than to isolated basigin. Certain invasion-inhibitory antibodies, that do not prevent binding of PfRH5 to isolated basigin, do in contrast prevent binding to the basigin complexes, explaining the mode of action of these previously enigmatic antibodies and providing valuable data towards the improved design of vaccines based on PfRH5.

## Introduction

Malaria is still one of the most deadly parasitic diseases to affect humans, with *Plasmodium falciparum* as the causative agent of the most severe cases (*Sato, 2021*). The clinical symptoms of malaria occur as the merozoite developmental form of the parasite invades and replicates within human red blood cells (RBCs) (*Venugopal et al., 2020*). Vaccines or therapeutics which block erythrocyte invasion therefore have the potential to contribute to reduction and elimination of malaria.

Erythrocyte invasion is driven by a series of molecular interactions between merozoite ligands and their receptors on erythrocyte surfaces (*Cowman et al., 2017*). While many of these interactions are redundant, the binding of merozoite *P. falciparum* reticulocyte homologue 5 (PfRH5) to erythrocyte basigin is essential for invasion by all tested *P. falciparum* strains (*Crosnier et al., 2011*), making PfRH5 one of the most promising blood-stage malaria vaccine candidates. PfRH5 is part of the five-component PfPCRCR complex, containing PfRH5, PfCyRPA, PfRIPR, PfPTRAMP, and PfCSS, and each component is essential for invasion and is a target of growth-inhibitory antibodies or nanobodies (*Farrell et al., 2023*; *Reddy et al., 2015*; *Scally et al., 2022*; *Wong et al., 2019*).

Vaccination of human volunteers with PfRH5 elicits strain-transcending anti-malarial antibodies (*Payne et al., 2017*) and, in a human challenge model, reduces the rate of parasite growth (*Minassian et al., 2021*). Structural studies of PfRH5 have revealed that basigin binds towards one of the tips of the 'kite-like' structure of PfRH5 (*Wright et al., 2014*). Structures of PfRH5 in complex with Fab fragments of monoclonal antibodies have identified a number of important epitopes across PfRH5, including those for human and mouse growth-inhibitory antibodies (R5.004, R5.016, 9AD4, and QA1) (*Alanine et al., 2019*; *Wright et al., 2014*), as well as the epitope for an antibody which potentiates the effects of growth-inhibitory antibodies (R5.011) (*Alanine et al., 2019*). This insight has guided development of improved PfRH5-containing vaccines (*Campeotto et al., 2017*). However, mysteries remain, with the most effective growth-inhibitory antibodies, 9AD4 and R5.016 not able to block basigin binding. How do they prevent invasion?

Basigin is a type I transmembrane protein of the immunoglobulin (Ig) superfamily, with the most common isoform presenting an ectodomain consisting of two highly glycosylated extracellular Ig domains (*Muramatsu, 2016*), both of which contact PfRH5 (*Wright et al., 2014*). More recently, the transmembrane segment of basigin has been shown to mediate formation of heteromeric complexes which contain basigin and either monocarboxylate transporters (MCTs) or plasma membrane calcium ATPases (PMCAs) (*Muramatsu, 2016*; *Supper et al., 2016*). MCTs are involved in proton-coupled exchange of lactate or pyruvate across plasma membranes (*Felmlee et al., 2020*). Both MCT1 and MCT4 interact with basigin in multiple eukaryotic cell lines (*Kirk et al., 2000*), affecting their function and stability (*Kirk et al., 2000*) and MCT1 is the primary subtype in human erythrocytes (*Juel et al., 2003*). PMCAs also play a role in membrane transport, with four subtypes of PMCA removing $Ca^{2+}$ from the cell cytosol to maintain intracellular $Ca^{2+}$ homeostasis and modulate cell signalling (*Stafford et al., 2017*). Either basigin, or its close paralog neuroplastin, are essential subunits of PMCA, influencing expression and function of PMCAs in rodent neurons, with human erythrocyte PMCAs complexed with basigin (*Schmidt et al., 2017*). Both basigin-MCT1 and neuroplastin-PMCA1 complexes have been structurally characterised, revealing interactions mediated by transmembrane helices (*Gong et al., 2018*; *Wang et al., 2021*), with the PfRH5 binding site on the basigin ectodomain remaining accessible.

Proteomics studies have confirmed that PMCA1, PMCA4, and MCT1 are all expressed on human RBC plasma membrane (*Ravenhill et al., 2019*) and, intriguingly, the actions of both MCTs and PMCAs have been linked to severe malaria (*Bedu-Addo et al., 2013*; *Mariga et al., 2014*; *Timmann et al., 2012*). In the case of MCT1, cerebral malaria is associated with increased lactate concentrations, which may damage the blood-brain barrier. However, it is hard to envisage a mechanism by which modulation of MCT1 function by PfRH5 during the rapid process of erythrocyte invasion could contribute to the symptoms of malaria. In contrast, PMCA polymorphisms have been linked to the development of severe malaria in a genome-wide association study, suggesting that calcium homeostasis across the erythrocyte membrane may affect the outcomes of malaria. Spikes of increased intra-erythrocytic calcium have been observed during the parasite invasion process and these are dependent on the interaction between PfRH5 and basigin (*Volz et al., 2016*; *Weiss et al., 2015*), raising the question of whether localised blockage of PMCA-mediated calcium efflux may cause these calcium spikes. In this study, we therefore asked whether PfRH5 can interact with basigin which is complexed with MCT1

and PMCAs, whether this modulates PMCA function and whether this explains the function of growth-inhibitory antibodies.

## Results

### Basigin in human erythrocyte is primarily found in complex with PMCAs and MCTs

We first aimed to determine whether basigin is mostly free when extracted from human erythrocyte membranes or is predominantly in complex with PMCAs or MCTs. We solubilised human erythrocyte ghosts using a mild detergent mixture of n-dodecyl-β-D-maltose and cholesterol hemisuccinate (DDM:CHS), fractionated this extract by size exclusion chromatography and used western blotting to assess where basigin, PMCAs, and MCT1 elute (*Figure 1a*). Probing individual fractions with a basigin-binding antibody revealed a broad band indicating the presence of heterogeneously glycosylated basigin. Fractions rich in basigin also showed strong signals for both PMCAs and MCT1, indicating that the majority of basigin co-migrates with PMCAs and MCT1 (*Figure 1a*). The three components predominantly eluted in a region of the migration profile between standards of mass 157 kDa and 200 kDa, compatible with the mass of detergent-solubilised basigin-MCT1 and basigin-PMCA complexes (*Figure 1a*). No basigin was found in later fractions in the region of the profile which would correspond to the mass of a single basigin in a detergent micelle.

To calibrate this experiment, we expressed full-length basigin in baculovirus-infected Sf9 cells. This was extracted and purified using either CHAPS or DDM:CHS. The DDM:CHS-extracted protein purified with other protein components, migrated as a broad peak on a size exclusion column and showed complex binding kinetics for PfRH5 (*Figure 1—figure supplement 1a, b*). In contrast, protein extracted with CHAPS appeared pure on a gel, showed an elution profile indicating it to be monodispersed on a size exclusion column and bound to PfRH5, as determined by surface plasmon resonance, with binding kinetics which matched those of basigin ectodomain (*Figure 1—figure supplement 1a, b*). We therefore concluded that insect cell expressed full-length basigin also associates with binding partners in Sf9 cells which co-purify in DDM:CHS but are removed using the harsher detergent CHAPS. To generate a sample suitable to calibrate our size exclusion experiment, we therefore purified full-length basigin after solubilisation with CHAPS and then detergent exchanged this sample into DDM:CHS, producing monomeric full-length basigin in a detergent micelle which matched that of our basigin-containing complexes purified from erythrocytes (*Figure 1—figure supplement 1a, b*). This full-length basigin eluted at ~11.2 ml from a size exclusion column, later than the basigin-containing complexes extracted from erythrocytes (*Figure 1a*). These data suggest that erythrocyte basigin is predominantly found in complexes which co-elute with MCT1 and PMCA.

To more closely investigate the association of basigin with PMCAs and MCT1, we performed antibody shift assays in two different settings. First, we used size exclusion chromatography to assess whether antibodies alter the mobility of basigin-MCT1 complexes (*Figure 1—figure supplement 2*). A basigin-rich solubilised ghost extract was applied to a size exclusion column either in the presence or absence of an MCT1-targeting monoclonal antibody and the resultant fractions were probed on western blots for MCT1, basigin, or a glycophorin C control. While there was no shift of glycophorin C, the addition of the MCT1-specific antibody caused a shift in elution profile of the majority of the MCT1 and ~53% of basigin, suggesting them to be in complex (*Figure 1—figure supplement 2*). However, ~47% of the basigin did not shift. This identifies pools of basigin which are found in complex with and without MCT1. Next, we performed antibody shift experiments on detergent-solubilised human erythrocyte membrane extracts resolved by 2D blue native PAGE (*Figure 1b*, *Figure 1—figure supplement 1c*). Western blotting these 2D gels and staining with basigin, PMCA, and MCT1 antibodies indicated two separate populations of basigin-containing complexes, one of which co-migrated with PMCA and one of which co-migrated with MCT1. Inclusion of an antibody which quantitatively binds to basigin shifted both populations to higher molecular weight (*Figure 1b*). Signals at the appropriate molecular weight for free basigin were not observed in either the presence or absence of the basigin-binding antibody (*Figure 1b*, *Figure 1—figure supplement 1c*), even upon prolonged exposure of the blots.

In parallel, we conducted a depletion experiment. PMCAs and MCT1 were sequentially depleted from solubilised ghosts. PMCAs were first depleted by affinity capture on a calmodulin (CaM) resin, which binds to a regulatory site on the cytoplasmic tails of PMCAs (*Stafford et al., 2017*), while a

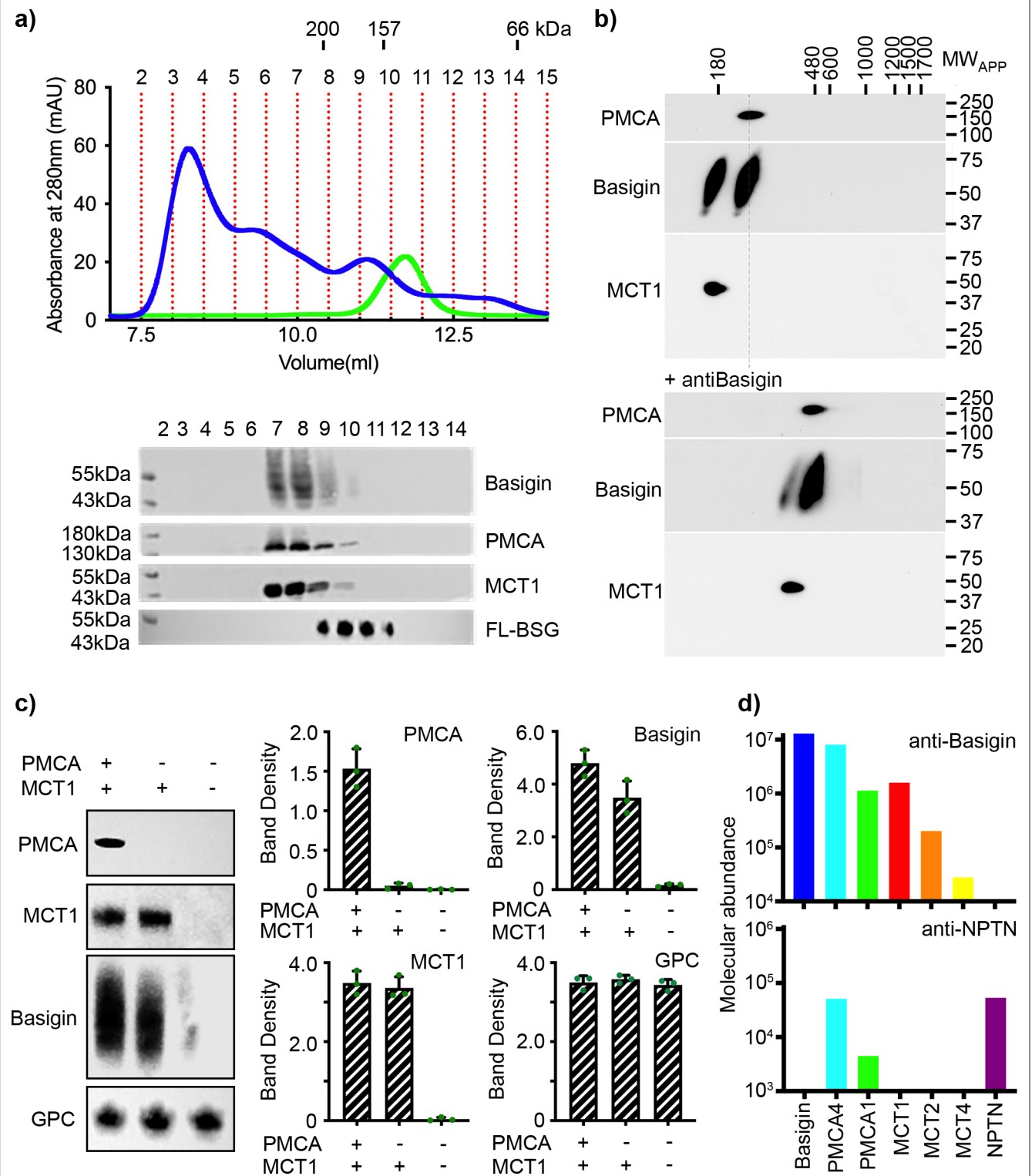

**Figure 1.** Basigin from human erythrocytes is found in complex with plasma membrane calcium ATPase (PMCA) or monocarboxylate transporter 1 (MCT1). (**a**) The upper panel shows the trace from size exclusion chromatography of *n*-dodecyl-β-D-maltose and cholesterol hemisuccinate (DDM/CHS)-solubilised ghost membrane proteins fractionated on a Superdex 200 increase 10/300 column (blue) and for full-length basigin purified from insect cells in CHAPS and exchanged into DDM:CHS (green). The elution volumes of molecular weight standards are indicated above the trace. Fractions collected are demarcated by vertical red dotted lines. The lower panel shows western blots of fractions 2–14. Representative blots for basigin (40–65 kDa, upper panel), PMCA (~138 kDa, upper middle panel), and MCT1 (~45 kDa, lower middle panel) show that all three proteins co-elute predominantly

*Figure 1 continued on next page*

*Figure 1 continued*

in fractions 7–9. The lower panel shows the equivalent blot for full-length basigin (FL-BSG), related to the green trace. Data shown are representative from n=2 biological replicates. (**b**) Western blot analysis of 2D blue native PAGE/SDS-PAGE separations of human erythrocyte membrane solubilisates before (upper panel) and after pre-incubation with anti-basigin antibody (lower panel). Blot membranes were stained with antibodies specific for PMCA1/4, basigin and MCT1. Markers of apparent complex size indicate the positions of known mitochondrial respiratory chain (super)complexes (*Schägger and Pfeiffer, 2000*) run in a separate gel lane. Binding of the antibody led to a full size-shift of both PMCA1/4-basigin and MCT1-basigin complexes, whereas no signal of free basigin could be observed in the low molecular weight range, even after overexposure of the blot. Data shown are representative from n=2 biological replicates. (**c**) The left-hand panel shows representative western blot images depicting sequential depletion of PMCA (upper panel) and MCT1 (upper-middle panel). Depletion of both transporters leads to reduced basigin levels, while levels of glycophorin C are unaffected in each fraction, again confirming that basigin is in complex with PMCA or MCT1. The remaining panels show densitometry plots obtained from inverted images of the western blots. Mean integrated band densities are shown with error bars as the standard error of the mean (n=3) and represent technical replicates. (**d**) Bar diagram depicting molecular abundances (abundance$_{norm}$spec values) of the indicated proteins in depleting affinity purifications with anti-basigin and anti-neuroplastin (NPTN) antibodies from mildly solubilised human erythrocyte membranes as determined by mass spectrometry. The abundance for all proteins was 0 after affinity purification with an IgG control. Data shown are representative from n=8 biological replicates for basigin and n=5 biological replicates for neuroplastin.

The online version of this article includes the following source data and figure supplement(s) for figure 1:

**Source data 1.** Data associated with *Figure 1*.

**Source data 2.** Gels and blots associated with *Figure 1*.

**Figure supplement 1.** Purification and biochemical characterisation of monomeric and complexed basigin.

**Figure supplement 1—source data 1.** Data associated with *Figure 1—figure supplement 1*.

**Figure supplement 1—source data 2.** Gels and blots associated with *Figure 1—figure supplement 1*.

**Figure supplement 2.** Mobility shift of basigin using a monocarboxylate transporter 1 (MCT1)-targeting antibody.

**Figure supplement 2—source data 1.** Data associated with *Figure 1—figure supplement 2*.

**Figure supplement 2—source data 2.** Gels and blots associated with *Figure 1—figure supplement 2*.

monoclonal antibody was used to subsequently immunodeplete MCT1. Western blotting was used to assess the fraction of PMCA, MCT1, and basigin depleted at each stage and these were quantified by densitometry (*Figure 1c*, *Figure 1—figure supplement 1d*). Sequential depletions of both PMCA and MCT1 resulted in almost complete elimination of basigin from the solubilised ghost membrane proteome, confirming that nearly all basigin is complexed with one of these transporters. Densitometric measurements also showed greater reduction in basigin levels upon MCT1 depletion, with ~72% of basigin complexed with MCT1 and ~28% of basigin complexed with PMCA. No basigin was depleted using beads alone (*Figure 1—figure supplement 1e*).

As there are multiple PMCA subtypes in human cells, we next used mass spectrometry analysis to identify which subtypes co-purify with basigin from human erythrocytes. This revealed PMCA4 to be the major partner of basigin in human erythrocytes, while also identifying PMCA1 (*Figure 1d*). Also identified were MCTs, with MCT1 as the primary subtype and MCT2 and -4 also detected (*Figure 1d*).

Collectively, these data show that native basigin in human erythrocytes is found only in heteromeric complexes, either with PMCA (predominantly PMCA4) or MCT1. As the interaction between PfRH5 and basigin has been shown to be essential for erythrocyte invasion by *P. falciparum*, this raises the question of whether PfRH5 can bind to either the basigin-PMCA and/or basigin-MCT1 complexes.

## PfRH5 binds to the basigin-PMCA complex with greater affinity than to monomeric basigin

We next investigated the binding of PfRH5 to basigin-PMCA complexes. The structure of the neuroplastin-PMCA1 complex shows the neuroplastin ectodomain to be presented above the extracellular loops of PMCA, with the surface of neuroplastin equivalent to that used by basigin to bind to PfRH5 being exposed, making it likely that PfRH5 can bind to PMCA-bound basigin (*Figure 2a*; *Gong et al., 2018*). To test this, we purified basigin-PMCA complexes from detergent solubilised human erythrocyte ghost cells using CaM affinity chromatography. Elution fractions showed a prominent band at ~138 kDa on a Coomassie-stained SDS-PAGE gel, corresponding to the mass of PMCA (*Figure 2—figure supplement 1a*). Western blotting confirmed the co-purification of basigin with PMCA (*Figure 2—figure supplement 1a*). Basigin and PMCA eluted as a single peak on a size exclusion column, with an elution profile matching a 200 kDa standard, suggesting complex formation

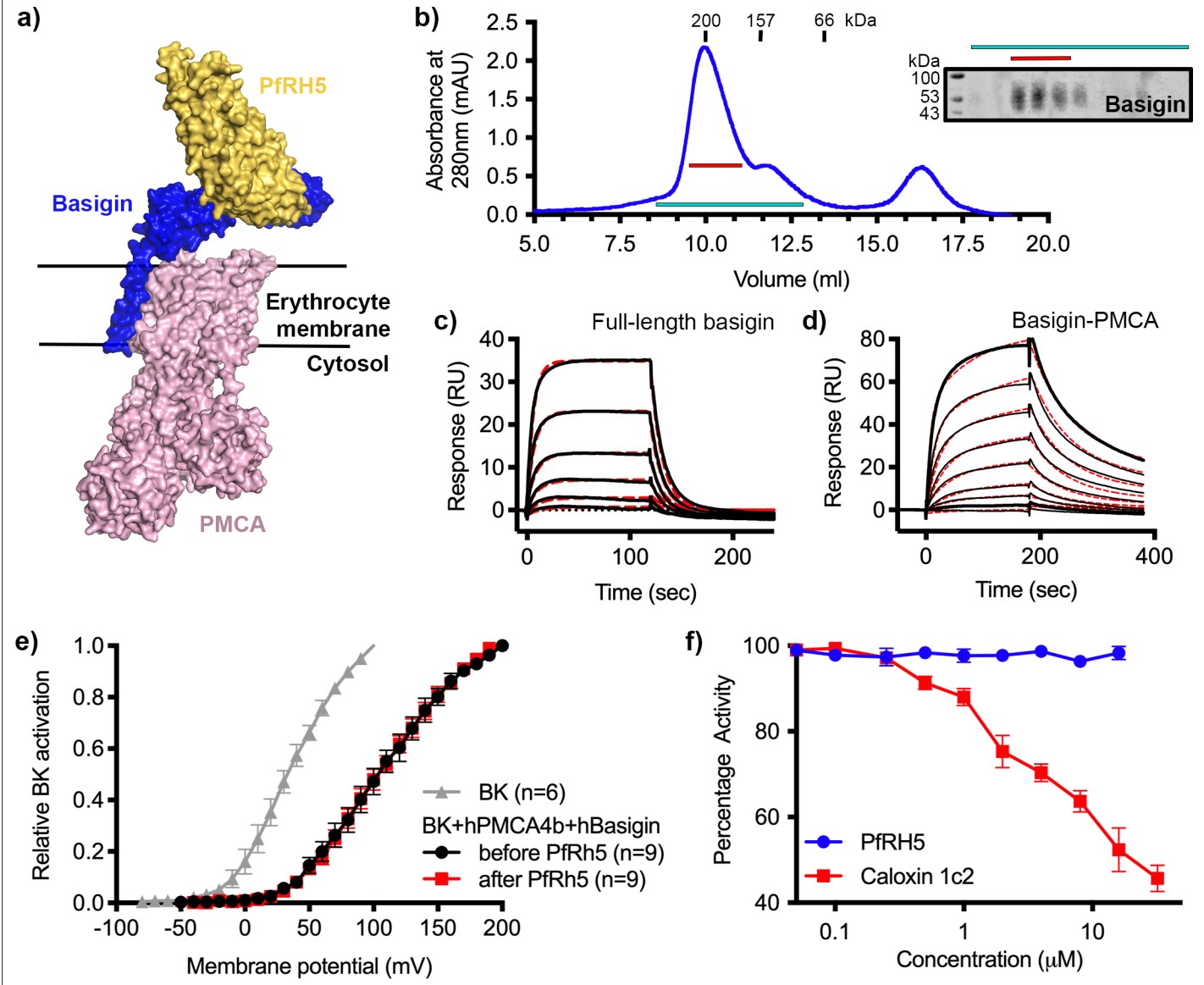

**Figure 2.** *P. falciparum* reticulocyte homologue 5 (PfRH5) binds to basigin-plasma membrane calcium ATPase (PMCA) without affecting calcium pumping activity. (**a**) A structural model showing a complex of basigin (blue) and PMCA (pink) (based on PDB: 6A69, *Gong et al., 2018*) onto which the complex of PfRH5 (yellow) and basigin (blue) (PDB:4U0Q, *Wright et al., 2014*) has been docked. (**b**) A size exclusion profile obtained for calmodulin affinity chromatography-purified PMCA from human erythrocyte ghosts separated on a Superdex 200 increase 10/300 column. The inset shows a western blot probed with an anti-basigin monoclonal antibody indicating co-migration of basigin with PMCA. Data shown are representative from n=3 biological replicates. (**c**) A surface plasmon resonance (SPR) sensogram showing the binding of a concentration series of full-length basigin (twofold dilutions from 4 µM) to immobilised PfRH5. Black lines show data and dashed red lines show fitting to a 1:1 binding model, with a dissociation constant of 0.56 µM. Data shown are representative from n=3 biological replicates. (**d**) An SPR sensogram showing the binding of a concentration series of basigin-PMCA (twofold dilutions from 400 nM) to immobilised PfRH5. Black lines show data and dashed red lines show fitting to a two-state binding model. The derived rate and affinity constants are presented in *Supplementary files 1 and 2*. Data shown are representative from n=2 biological replicates. (**e**) Activation curves of $BK_{Ca}$ channels recorded in Chinese hamster ovary (CHO) cells expressing $BK_{Ca}$ channels alone (grey), or cells also transfected with human PMCA4b and basigin with (red) and without (black) addition of PfRH5 at 2 µM concentration. Data shown are from n=3 biological replicates and error bars show standard error of mean. (**f**) Concentration-response curves for the inhibition of $Ca^{2+}$-ATPase activity, as determined by measuring inorganic phosphate production, of purified basigin-PMCA complex by known inhibitor caloxin 1c2 (red) and PfRH5 (blue). Each data point represents the mean and error bars represent the standard error of the mean of technical replicates (n=3, technical replicates).

The online version of this article includes the following source data and figure supplement(s) for figure 2:

**Source data 1.** Data associated with *Figure 2*.

**Source data 2.** Gels and blots associated with *Figure 2*.

*Figure 2 continued on next page*

*Figure 2 continued*

**Figure supplement 1.** Purification and functional characterisation of basigin-plasma membrane calcium ATPase (PMCA).

**Figure supplement 1—source data 1.** Data associated with *Figure 2—figure supplement 1*.

**Figure supplement 1—source data 2.** Gels and blots associated with *Figure 2—figure supplement 1*.

**Figure supplement 2.** Electrophysiology to assess basigin-plasma membrane calcium ATPase (PMCA) function.

**Figure supplement 2—source data 1.** Data associated with *Figure 2—figure supplement 2*.

**Figure supplement 2—source data 2.** Gels and blots associated with *Figure 2—figure supplement 2*.

(*Figure 2b*). The purified complex showed $Ca^{2+}$-dependent ATP hydrolysis activity, confirming that active basigin-PMCA complexes had been isolated (*Figure 2—figure supplement 1b*).

Binding of purified basigin-PMCA complexes to PfRH5 was measured by surface plasmon resonance (SPR) analysis. PfRH5 was chemically biotinylated on lysine residues and was immobilised on a streptavidin-coated chip and either 0.4 μM basigin-PMCA or 4 μM of basigin ectodomain were injected. Basigin ectodomain bound with similar kinetics to those previously observed (*Figure 2—figure supplement 1c*; *Crosnier et al., 2011*; *Wright et al., 2014*). Basigin-PMCA complexes bound PfRH5 with a sensogram profile indicating slower on- and off-rates (*Figure 2—figure supplement 1d*). To confirm that these responses are specific, we pre-incubated surface-immobilised PfRH5 with monoclonal antibody R5.004, which occludes the basigin binding site (*Alanine et al., 2019*). This blocked the binding of both basigin ectodomain and basigin-PMCA to PfRH5, as expected (*Figure 2—figure supplement 1c and d*).

To measure binding kinetics, we produced PfRH5 carrying an N-terminal biotin acceptor peptide (BAP) tag. This was captured at low density on a streptavidin-coated SPR chip and binding was studied by injecting twofold dilution series of either basigin ectodomain (*Figure 2—figure supplement 1e*), full-length basigin (purified in CHAPS and exchanged into DDM:CHS) (*Figure 2c*), or basigin-PMCA (*Figure 2d*). Kinetic binding data for full-length basigin could be fitted to a 1:1 binding model with an affinity of 0.56 μM ($x^2$=2.44), in good agreement with the affinity of ~1 μM measured for basigin ectodomain in previous studies (*Figure 2—figure supplement 1e*; *Crosnier et al., 2011*; *Wright et al., 2014*). In contrast, the kinetic data for the basigin-PMCA complexes did not fit well to a 1:1 binding model ($x^2$=7.71) (*Figure 2—figure supplement 1f*), but the fit improved significantly ($x^2$=1.85) when using a two-state binding model (*Figure 2—figure supplement 1f*, *Supplementary files 1 and 2*). This model is consistent with an interaction which involves two separate, successive events, such as an initial lower affinity capture event, followed by a second binding event which increases the overall affinity, or a conformational change which follows an initial binding event. PfRH5 therefore bound to basigin-PMCA ~9-fold more tightly than to full-length basigin, primarily due to a lower dissociation rate constant (*Supplementary files 1 and 2*).

## PfRH5 binding does not modulate calcium transport mediated by basigin-PMCA complexes

We next investigated whether PfRH5 binding can modulate the activity of PMCA, perhaps accounting for changes in calcium flux at the junction between the erythrocyte and invading merozoite. To test this, we used an established system in which current recordings from $BK_{Ca}$-type $Ca^{2+}$-activated $K^+$ channels were used as a readout of apparent intracellular $Ca^{2+}$ concentration (*Schmidt et al., 2017*). The expression of PMCA4b and basigin, together with $BK_{Ca}$, in this system caused reduced intracellular $Ca^{2+}$ concentrations, as demonstrated by a right shift in the $BK_{Ca}$ activation curve. We confirmed that the PMCA4-basigin complex formed in these cells using 2D blue native PAGE western blot analysis of solubilised membranes (*Figure 2—figure supplement 2a*). We next added PfRH5 at a concentration of 2 μM which caused no further change in the $BK_{Ca}$ activation curve, indicating that PfRH5 binding did not modulate PMCA-basigin function and did not change the intracellular $Ca^{2+}$ concentration (*Figure 2e*). As PfRH5 is normally part of the three-component PfRCR complex, we also added PfRCR at a concentration of 1 μM to the same system. Here too, we observed no effect on intracellular $Ca^{2+}$ concentration (*Figure 2—figure supplement 2b and c*). Finally, we studied the effect of PfRH5 on $Ca^{2+}$-dependent ATPase activity of purified basigin-PMCA by following inorganic phosphate release through a colorimetric assay. While the extracellular peptide inhibitor caloxin 1c2 inhibited calcium-dependent ATP hydrolysis at high micromolar concentrations, the addition of PfRH5 did not alter the

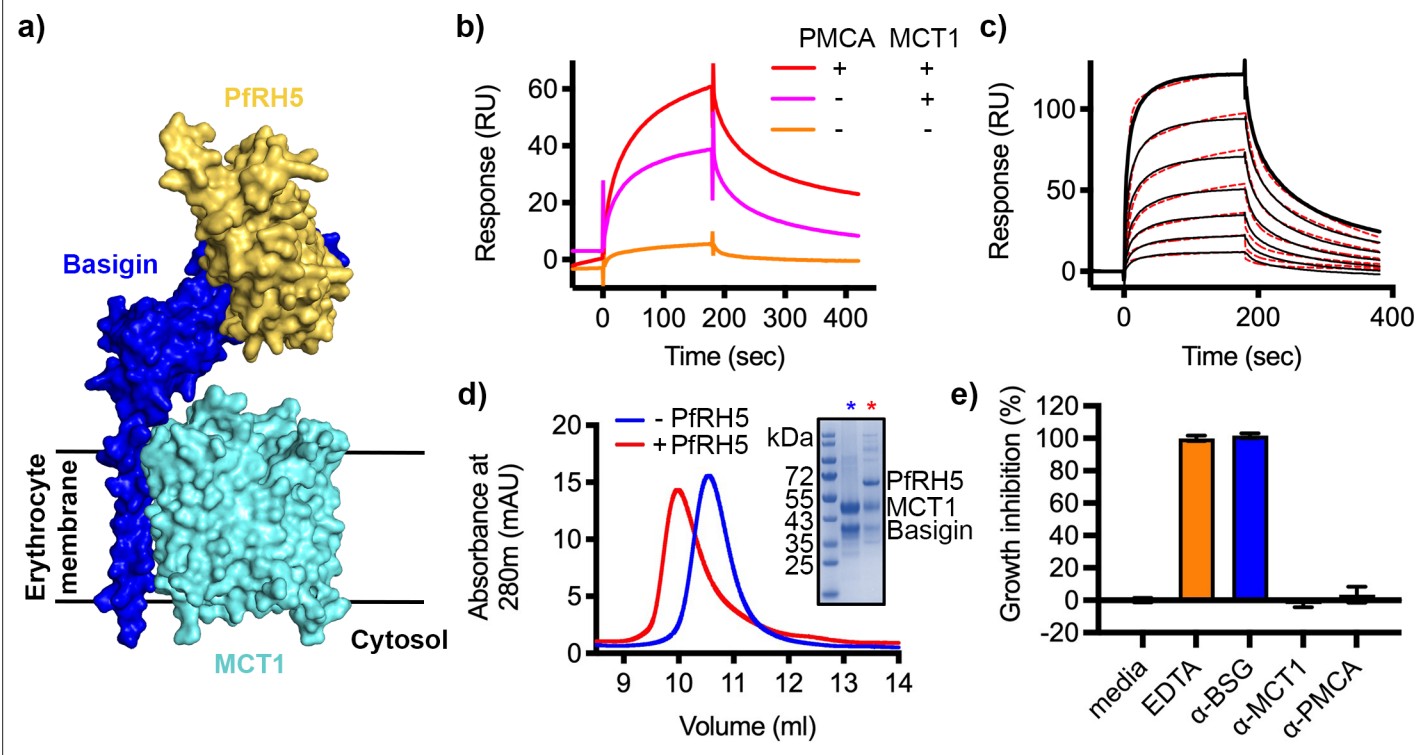

**Figure 3.** *P. falciparum* reticulocyte homologue 5 (PfRH5) binds to basigin-monocarboxylate transporter 1 (MCT1). (**a**) A structural model showing a complex of basigin (blue) and MCT1 (cyan) (based on PDB: 6LYY, *Wang et al., 2021*) onto which the complex of PfRH5 (yellow) and basigin (blue) (PDB:4U0Q, *Wright et al., 2014*) has been docked. (**b**) Surface plasmon resonance (SPR) traces after flowing detergent solubilised erythrocyte membrane basigin-rich fractions (red) and membrane fractions depleted for plasma membrane calcium ATPase (PMCA) (pink) and both PMCA and MCT1 (orange) over a PfRH5-coated surface. (**c**) An SPR sensogram showing the binding of a concentration series of basigin-MCT1 (twofold dilutions from 1600 nM) to immobilised PfRH5. Black lines show data and dotted red lines show fitting to a two-state binding model. Data shown are representative from n=3 technical replicates. (**d**) Purified PfRH5 and basigin-MCT1 were assayed for complex formation through size exclusion chromatography using a Superdex 200 increase 10/300 column. The elution profile of basigin-MCT1 alone (blue) and in the presence of PfRH5 (red) are shown. The inset SDS-PAGE gel shows the protein species present in the fractions indicated by stars in the elution trace. (**e**) Growth inhibition assays assessed the effect of antibodies targeting basigin (at 1 µg/ml) and MCT-1 and PMCAs (both at 75 µg/ml) on *P. falciparum* growth in human blood culture. Five mM EDTA was used as a positive control (100% growth inhibition) while complete media was used as a negative control (0% growth inhibition). Data are the mean and standard deviation of three technical replicates, with a single example from three biological repeats shown.

The online version of this article includes the following source data and figure supplement(s) for figure 3:

**Source data 1.** Data associated with *Figure 3*.

**Source data 2.** Gels and blots associated with *Figure 3*.

**Figure supplement 1.** Surface plasmon resonance analysis of binding of *P. falciparum* reticulocyte homologue 5 (PfRH5) to basigin-monocarboxylate transporter 1 (MCT1).

**Figure supplement 1—source data 1.** Data associated with *Figure 3—figure supplement 1*.

**Figure supplement 2.** Assessment of antibodies targeting basigin, plasma membrane calcium ATPase (PMCA), and monocarboxylate transporter 1 (MCT1).

**Figure supplement 2—source data 1.** Gels and blots associated with *Figure 3—figure supplement 2*.

ATPase activity of PMCA (*Figure 2f*). We therefore see no evidence that PfRH5 binding affects the activity of the basigin-PMCA calcium pump.

## PfRH5 binds to the basigin-MCT1 complex with greater affinity than for monomeric basigin

As a large fraction of erythrocyte basigin is associated with MCT1, we also tested whether the basigin-MCT1 complex can bind to PfRH5, as the PfRH5-binding site of basigin is exposed in the structure of the basigin-MCT1 complex (*Wang et al., 2021*; *Figure 3a*). We first studied the binding of PfRH5

to solubilised ghosts and their depleted forms using SPR. Flowing solubilised ghost membrane extract over a chip surface coated with PfRH5 produced a binding response (*Figure 3b*). This reduced ~22% after depletion of PMCA-containing complexes, and the binding response was nearly absent for samples depleted for both PMCAs and MCT1 (*Figure 3b*). This suggests that basigin in complex with either MCT1 or PMCAs can bind to PfRH5.

We next expressed the basigin-MCT1 complex in Sf9 cells and tested its binding to PfRH5. To measure the affinity of the interaction, monobiotinylated PfRH5 was coupled to an SPR chip before injection of a twofold dilution series of purified basigin-MCT1 complex. Again, the kinetic data could be best described by a two-state binding model ($\chi^2$=3.09) with a $K_D$ value of 93 nM (*Figure 3c*, *Figure 3—figure supplement 1a*, *Supplementary files 1 and 2*). One of the predictions of a two-state binding model is that longer periods of incubation of PfRH5 with the basigin-MCT1 complex allow a larger fraction of interacting species to form stable complexes due to completion of the second binding event, and lead to slower off-rates. When we injected basigin-MCT1 over a chip decorated with biotinylated PfRH5, we indeed observed slower off-rates to be associated with longer incubation times, confirming a two-state interaction (*Figure 3—figure supplement 1b*). We also mixed basigin-MCT1 with PfRH5 for analysis by size exclusion chromatography, and again observed formation of a stable complex (*Figure 3d*). Therefore, PfRH5 binds to both basigin-PMCA and basigin-MCT1 with similar affinities, both more tightly than the interaction with the isolated basigin ectodomain.

## Two antibodies that bind to PMCAs and MCT1 do not inhibit parasite growth

We next assessed whether polyclonal antibodies that bind to PMCAs and MCT1 can prevent the growth of *P. falciparum* in human blood culture. We obtained the only two commercially available antibodies which bind to the extracellular regions of either PMCAs or MCT1, together with the growth-inhibitory basigin-binding monoclonal antibody, TRA-1–85. We first assessed the ability of these antibodies to stain human erythrocytes by immunofluorescence imaging, confirming that they bind to basigin-MCT1 and basigin-PMCA complexes in the context of intact erythrocytes (*Figure 3—figure supplement 2*). We next conducted a standard one-cycle growth inhibition assay which assessed the ability of these antibodies to block growth of *P. falciparum* in human blood culture (*Figure 3e*). As previously shown, the basigin-targeting antibody, TRA-1–85, showed complete inhibition of growth at a 1 µg/ml concentration (*Crosnier et al., 2011*). However, neither the PMCA or MCT1 antibodies caused growth inhibition at 75 µg/ml concentration. In neither case are structures available to show how these antibodies bind to their receptors, making it impossible rationalise why they were not inhibitory. We envisage that it may be possible to raise growth-inhibitory antibodies which target MCT1 or PMCAs, but with strongly growth-inhibitory monoclonal antibodies that target PfRH5 already available, antibodies targeting MCT1 or PMCAs are unlikely to be therapeutically valuable.

## Growth-inhibitory antibodies sterically block binding of PfRH5 to basigin-PMCA and basigin-MCT1 complexes

While antibodies which occlude the basigin binding site of PfRH5 are growth-inhibitory, some of the most potent PfRH5-binding growth-inhibitory antibodies do not prevent PfRH5 from binding to basigin, raising questions about their mode of action. We therefore tested the ability of a panel of structurally characterised antibodies to prevent PfRH5 from binding to basigin ectodomain, full-length basigin, basigin-PMCA, and basigin-MCT1. The panel consisted of one growth-inhibitory antibody, R5.004, which prevents basigin binding, and two potent growth-inhibitory antibodies, R5.016 and 9AD4, which do not prevent basigin binding. Also tested were non-inhibitory antibodies, R5.011 and R5.015, which do not block basigin binding.

In accordance with previously published findings, only R5.004 prevented the binding of basigin ectodomain to PfRH5 immobilised on an SPR chip (*Figure 4—figure supplement 1a*). A similar outcome was seen for full-length basigin, showing that the presence of the detergent micelle is not sufficient to allow R5.016 to cause steric occlusion of binding (*Figure 4a*, *Figure 4—figure supplement 1b*). In contrast, both R5.016 and 9AD4 reduced binding of basigin-PMCA to immobilised PfRH5 by >90% (*Figure 4b*, *Figure 4—figure supplement 1b*). Indeed R5.016 inhibited binding of basigin-PMCA to PfRH5 in a concentration-dependent manner, with concentrations of 100 nM or above producing nearly 90% inhibition (*Figure 4—figure supplement 1c and d*). Both R5.016 and

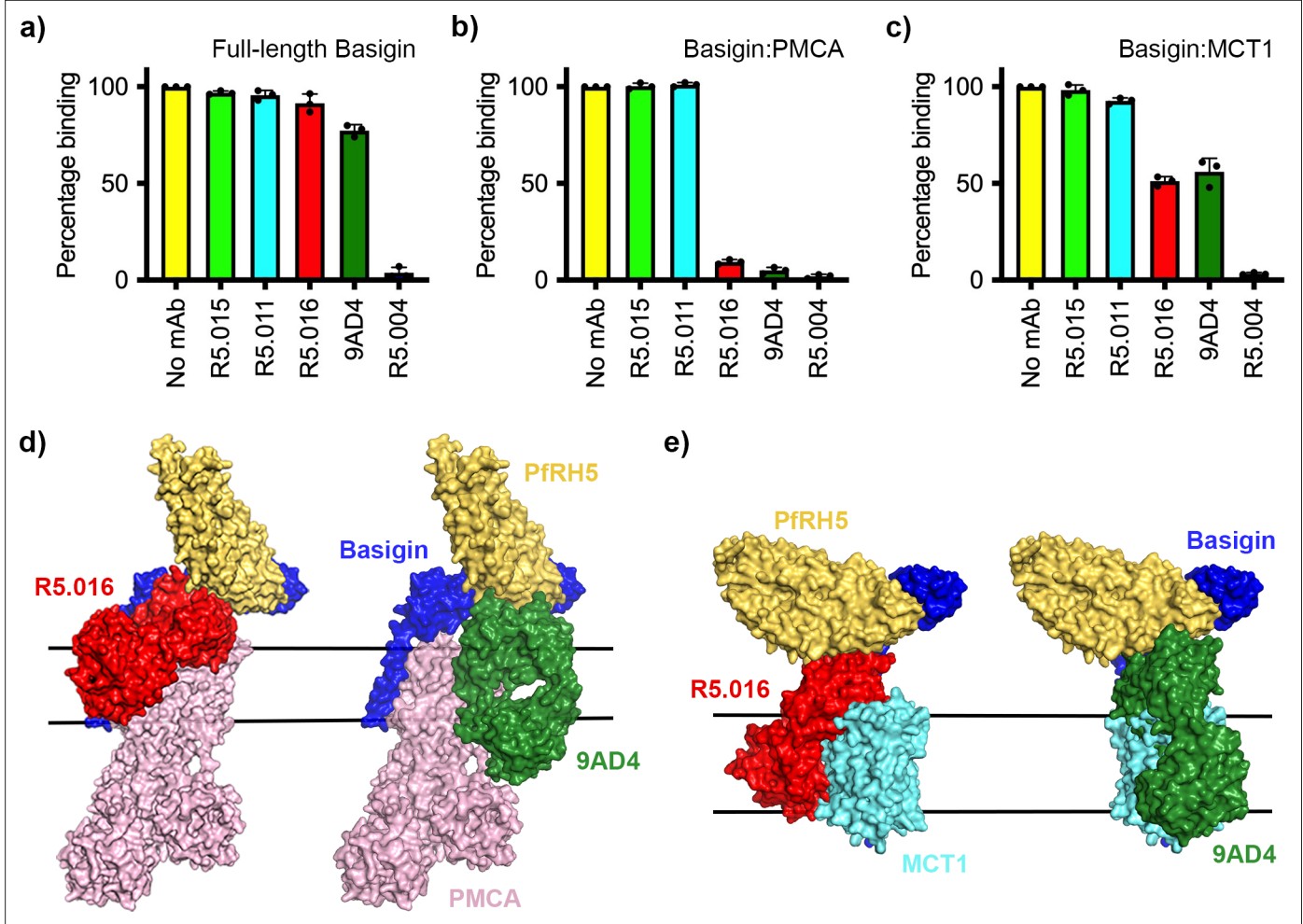

**Figure 4.** Neutralising monoclonal antibodies targeting *P. falciparum* reticulocyte homologue 5 (PfRH5) prevent its binding to basigin-plasma membrane calcium ATPase (PMCA) and basigin-monocarboxylate transporter 1 (MCT1) complexes. Five different PfRH5-binding monoclonal antibodies were tested for the inhibition of the binding of PfRH5 to (**a**) full-length basigin, (**b**) basigin-PMCA complex, and (**c**) basigin-MCT1 complex. In each case data (n=3) shown are the mean and error bars represent standard error of mean of biological replicates for full-length basigin and basigin-PMCA and technical replicates for basigin-MCT1. (**d**) Structural models showing a complex of basigin (blue) and PMCA (pink) (based on PDB: 6A69, *Gong et al., 2018*) onto which the complex of PfRH5 (yellow) and basigin (blue) (PDB:4U0Q, *Wright et al., 2014*) has been docked. Onto this model has been docked either the complex of PfRH5 (yellow) bound to the Fab fragment of R5.016 (red, PDB:6RCV, *Alanine et al., 2019*) or of 9AD4 (green; 4U0R, *Wright et al., 2014*). (**e**) Structural models showing a complex of basigin (blue) and MCT1 (cyan) (based on PDB: 6LYY, *Wang et al., 2021*) onto which the complex of PfRH5 (yellow) and basigin (blue) (PDB:4U0Q, *Wright et al., 2014*) has been docked. Onto this model has been docked either the complex of PfRH5 (yellow) bound to the Fab fragment of R5.016 (red, PDB:6RCV, *Alanine et al., 2019*) or of 9AD4 (green; 4U0R, *Wright et al., 2014*).

The online version of this article includes the following source data and figure supplement(s) for figure 4:

**Source data 1.** Data associated with *Figure 4*.

**Figure supplement 1.** Neutralising monoclonal antibodies targeting *P. falciparum* reticulocyte homologue 5 (PfRH5) prevent its binding to basigin-plasma membrane calcium ATPase (PMCA) and basigin-monocarboxylate transporter 1 (MCT1) complexes.

**Figure supplement 1—source data 1.** Data associated with *Figure 4—figure supplement 1*.

9AD4 also reduced the binding of the basigin-MCT1 complex to immobilised PfRH5, with 100 nM of either antibody reducing binding by more than 50% (*Figure 4c*, *Figure 4—figure supplement 1b*). In contrast, neither of the non-inhibitory antibodies tested reduced the binding of basigin-PMCA or basigin-MCT1 to PfRH5.

To rationalise the mechanism of inhibition, we docked crystal structures of PfRH5 complexed with basigin (*Wright et al., 2014*), R5.016 (*Alanine et al., 2019*), and 9AD4 (*Wright et al., 2014*) Fab fragments onto the neuroplastin-PMCA structure (*Gong et al., 2018*; *Figure 4d*) and the basigin-MCT1

structure (*Wang et al., 2021*; *Figure 4e*) to generate composite models. In both cases, binding of R5.016 and 9AD4 to PfRH5 would physically occlude its binding to basigin in the basigin-PMCA or basigin-MCT1 complexes, either through direct clashes with MCT1 or PMCA, or through a clash with the plasma membrane. Indeed, these structural models are constructed using only the Fab fragments of the monoclonal antibodies and, in the presence of the antibody constant domains, even greater steric clash would be expected. These data therefore support a model in which neutralising antibodies prevent PfRH5 from binding to basigin in the context of macromolecular basigin-containing complexes.

## Discussion

Numerous studies have now confirmed that the interaction between PfRH5 and basigin is required for invasion of *P. falciparum* into human erythrocytes (*Crosnier et al., 2011*). To date, no strain of *P. falciparum* has been identified which can take a basigin-independent invasion route. Blocking this interaction also has proven therapeutic potential to prevent malaria, with antibodies against both PfRH5 and basigin identified which inhibit invasion (*Alanine et al., 2019*; *Douglas et al., 2014*; *Wright et al., 2014*; *Zenonos et al., 2015*), with vaccination (*Douglas et al., 2015*) or passive transfer of neutralising antibodies (*Douglas et al., 2019*) leading to protection of Aotus monkeys from parasite challenge and with human vaccination slowing the onset of parasitaemia (*Minassian et al., 2021*). Nevertheless, mysteries remain. First, the functional consequences of the PfRH5-basigin interaction are still unclear. What are the downstream outcomes of the interaction and why are they essential for invasion? Second, some of the most potent human and mouse PfRH5-targeting monoclonal antibodies which block parasite invasion do not directly block the PfRH5-basigin interaction. How do they function?

To answer these questions, we started by investigating the immediate molecular context of basigin in human erythrocytes. Various studies in different cell types have shown that basigin forms heteromeric complexes with either PMCAs or MCTs (*Muramatsu, 2016*). We therefore used a combination of depletion, purification, 2D blue native PAGE, and mobility shift experiments to show that nearly all basigin in human erythrocyte membranes is found in complex with either PMCAs or MCTs (*Figure 1*). While ~20% of basigin was not depleted with either PMCAs or MCT1 (*Figure 1c*), all basigin migrated at a high molecular weight on a size exclusion column and we did not observe free basigin in 2D blue native PAGE experiments. Indeed, techniques that more carefully preserve complex formation showed all basigin to be associated with PMCAs or MCTs (*Figure 1b*, *Figure 1—figure supplement 1c*). The detection of free basigin in depletion experiments may then be due to disruption of protein complexes during depletion experiments, or due to basigin in complex with subtypes or splice isoforms of PMCA or other subtypes of MCTs that are not efficiently depleted. Therefore, nearly all basigin in human erythrocytes is found within larger protein complexes, with consequences for its recognition by PfRH5 and its role in *Plasmodium* invasion.

Intriguingly, both MCT1- and PMCA-bound basigin bind more tightly to PfRH5 than to isolated full-length basigin. Indeed, fitting of kinetic binding data from SPR showed that, while the binding of PfRH5 to basigin fits a simple 1:1 binding mode, its binding to basigin-PMCA and basigin-MCT1 shows biphasic kinetics, which fit most closely to a two-state binding model. This is further supported by data which shows that PfRH5 binds to basigin-MCT1 such that increasing association times correlate with decreasing dissociation rates, suggesting an initial lower affinity interaction, which 'matures' into a higher affinity interaction over time. While kinetic fitting data cannot be used to strictly derive a molecular mechanism, this is compatible with an initial lower affinity interaction between PfRH5 and basigin, followed by a subsequent interaction which stabilises the complex, possibly between PfRH5 and another part of the basigin-PMCA or basigin-MCT1 complexes.

Our demonstration that PfRH5 can bind to basigin-PMCA and basigin-MCT1 next led us to investigate whether PfRH5 binding can modulate the functions of these two transporters. MCT1 is a lactate transporter, and we were not able to envisage a mechanism by which altered lactate transport could play an essential role in the context of the ~20-s-long process of erythrocyte invasion. However, PfRH5 is known to be essential for a spike of increased calcium concentration at the merozoite-erythrocyte junction (*Volz et al., 2016*; *Weiss et al., 2015*) and the presence of basigin has been shown to alter PMCA function (*Schmidt et al., 2017*). Could modulation of basigin-PMCA function block calcium efflux and cause this spike during erythrocyte invasion? Our data do not support this hypothesis,

with no change in basigin-PMCA function observed, either in the presence of PfRH5 or of the PfRCR complex. How PfRH5 modulates erythrocyte calcium levels to facilitate invasion is therefore still unclear. However, the demonstration that erythrocyte basigin is primarily found in large multimeric complexes will provide clues for future research to unveil this mechanism.

Finally, our results reveal the mechanisms by which the most potent PfRH5-targeting neutralising antibodies can block invasion. Immunity to blood-stage malaria is antibody-mediated and, while antibodies which target other aspects of the malaria life cycle use Fc-dependent mechanisms of immunity (*Kurtovic et al., 2020*; *Teo et al., 2016*), those which prevent invasion appear to function directly by blocking essential interactions (*Alanine et al., 2019*; *Douglas et al., 2019*; *Wright et al., 2014*). As a result, the outcome of in vitro growth inhibition assays correlates well with protection due to PfRH5-targeting antibodies (*Douglas et al., 2019*). It was therefore a mystery why the most effective PfRH5-targeting neutralising antibodies from both humans (R5.016) (*Alanine et al., 2019*) and mice (9AD4) (*Douglas et al., 2014*; *Wright et al., 2014*) do not interfere with the PfRH5-basigin interaction. Studying basigin in the context of its membrane protein binding partners provides an answer, with both R5.016 and 9AD4 inhibiting the binding of PfRH5 to basigin-PMCA and basigin-MCT1 complexes. While this inhibition is strong in the context of a detergent micelle, we would expect it to be even stronger in the membrane context, with both R5.016 and 9AD4 modelled to angle directly towards the membrane plane. Our studies therefore highlight the importance of understanding the molecular context of basigin in understanding PfRH5 function and its inhibition, and open new directions to understand why the PfRH5-basigin interaction is so crucial for erythrocyte invasion.

## Materials and methods
### Ghost preparation from human erythrocytes
400–450 ml of packed human RBCs were purchased from the NHS blood transfusion service after ethical consideration from the relevant University of Oxford ethics committee. No information about the age, sex, gender, and ethnicity of the donors was available. RBCs were lysed in 10 volumes of hypotonic buffer (10 mM Tris-Cl, pH 8.0 and 1 mM EDTA and 1 mM PMSF) at 4°C for 2 hr. The lysate was first centrifuged at 1500 × $g$ for 30 min to remove cellular debris and then circulated through four omega T-series centramate 0.1 sq. ft tangential filter flow unit (100 kDa, PALL) to generate 300–350 ml of concentrated retentate of RBC membranes. The retentate was spun at 100,000 × $g$ for 45 min at 4°C and bulk haemoglobin was very carefully pipetted from the top without disturbing the pellet. To remove residual haemoglobin, retrieved membrane pellets were resuspended in hypotonic buffer and centrifuged at 100,000 × $g$ for 20 min at 4°C. This process was repeated at least four times or until white to light pink ghosts were obtained. Ghosts corresponding to 100 ml of packed RBCs were stored in 50 ml of buffer containing 20 mM HEPES-Na, pH 7.2, 150 mM NaCl, 10% glycerol supplemented with complete protease inhibitor cocktail EDTA-free (Roche) at –80°C until further use.

### Fractionation and analysis of solubilised ghosts by size exclusion chromatography
0.5 ml of ghost membranes were solubilised by adding 10% of a mixture of DDM:CHS in a 10:1 wt/wt ratio (DDM-CHS) solution to final concentration of 1.2% and incubating for 1 hr at 4°C with gentle stirring. Solubilised membranes were spun at 100,000 × $g$ for 45 min at 4°C, and the supernatant was injected into a Superdex 200 increase 10/300 GL (cytiva) column calibrated with known molecular weight standards (GE Healthcare). Size exclusion chromatography was then performed at a flow rate of 0.5 ml/min in a buffer containing 20 mM HEPES, pH 7.2, 150 mM NaCl, and 0.02% DDM/CHS where proteins eluting from the column were collected as discrete fractions of 500 µl each. 10–15 µg protein from each fraction was separated on a 10% SDS-PAGE gel and blotted on a PVDF membrane using Bio-Rad turbo transfer pack and apparatus. Membranes were blocked with 6% TBS-milk subsequently probed with 1:500 mouse anti-human PMCA IgG (5F3, Santa Cruz Biotechnology), 1:1000 mouse anti-human CD147 IgG (Biolegend), and 1:1000 mouse anti-human MCT1 IgG (Santa Cruz Biotechnology) overnight at 4°C. Washes were conducted in TBS and TBS-T-20 0.05% and the membrane was then incubated with anti-mouse IgG 1:20,000 for 45 min at room temperature. After final washes, blots were developed and visualised using ECL reagent SuperSignal West Atto (Thermo Scientific) and i-Bright imager (Life Technologies).

### Mobility shift using an antibody against MCT1

200 µl of 0.1–0.15 mg of fractions of DDM-CHS solubilised human ghost proteins rich in basigin were incubated with 20 µg of mouse anti-human MCT1 IgG and incubated for 30 min on ice. Following the incubation, the sample was injected into a Superdex 200 increase 10/300 GL column and eluted at 0.5 ml/min as fractions of 500 µl. 10–15 µg protein from each SEC fraction from samples with or without MCT1 antibody were separated by 12% SDS-PAGE, prior to western blotting. Detection of MCT1 and basigin was performed on separate blots, using 1:1000 of polyclonal rabbit anti-human MCT1 IgG (Cell Signaling) and 1:1000 mouse anti-human CD147 IgG (Biolegend) followed by HRP-conjugated secondary IgG antibody of specific host type. Proteins were visualised with ECL reagent SuperSignal West Atto using i-Bright imager (Life Technologies).

### Depletion of native PMCA and MCT1

0.20 mg of solubilised ghost proteins were incubated with 30 µl of CaM resin for 45 min at 4°C to deplete native PMCA. Beads were removed by centrifugation at 250 × $g$ for 5 min. 50 µg was set aside for analysis and the remaining sample was mixed with 10 µg of mouse anti-human MCT1 IgG and incubated for 30 min on ice. Following the incubation, 50 µl of protein G resin was added to the mixture and was further incubated for 2 hr on a rotating platform in a cold room. Supernatant from this depleted mixture was collected by centrifugation to remove beads. 10–15 µg of protein from solubilised ghost and each depleted fraction was separated on 12% SDS-PAGE for side-by-side analysis using western blotting. Blots were developed and visualised as described above.

### Affinity purification from human erythrocytes for mass spectrometry analysis

Membranes were prepared from human erythrocytes by hypotonic lysis (1:50 (vol:vol) in 1 mM EDTA/EGTA, 10 mM Tris-HCl, pH 7.4+protease inhibitors) followed by ultrasound treatment (2×20 pulses, duty cycle 50, output 1 (Branson Sonifier 250)) and ultracentrifugation for 10 min at 150,000 × $g$. Membrane pellets were resuspended at 11 mg/ml (as determined by Bradford assay, Bio-Rad) in 20 mM Tris-HCl, pH 7.4. Then 2 mg of membrane were solubilised in 2 ml ComplexioLyte 47 ready-to-use detergent buffer (Logopharm) supplemented with 1 mM EDTA/EGTA and protease inhibitors. After ultracentrifugation for 10 min at 250,000 × $g$, the solubilisate was precleared by incubation with 15 µg bead-immobilised control IgG (Millipore 12–370) for 1 hr at 4°C. Two 0.5 ml aliquots of the supernatant were then mixed with 5 µg of bead-immobilised anti-basigin (Origene TA501189) and anti-neuroplastin (R&D Systems AF7818) antibodies, respectively. After 2 hr incubation at 4°C, beads were washed twice for 10 min in ComplexioLyte 47 washing buffer (Logopharm) and eluted with 2×10 µl of denaturing buffer (non-reducing Laemmli buffer+8 M urea, 2×10 min at 37°C). The eluates were then supplemented with 100 mM DTT, run on an SDS-PAGE gel and silver-stained. Lanes were split into high- and low-MW range sections and subjected to standard in-gel tryptic digestion for subsequent LC-MS/MS analysis. Target depletion in these affinity purifications was verified by western blot analysis (not shown).

### Mass spectrometry and protein quantification

LC-MS/MS analysis was carried out in positive ion mode on an LTQ Orbitrap XL mass spectrometer (Thermo Scientific, Germany; CID fragmentation of the five most abundant new precursors per scan cycle; singly charged ions rejected; dynamic exclusion duration 30 s) equipped with a split-based Ulti-Mate 3000 HPLC (Dionex/Thermo Scientific, Germany; 'short' gradient) as detailed in *Schmidt et al., 2017*. Peak lists were extracted from MS/MS spectra using 'msconvert.exe' (ProteoWizard; https://proteowizard.sourceforge.io/; v3.0.6906). After an initial database search precursor mass (m/z) values were corrected by linear shifting of their median offset. Final searches against the UniProtKB/Swiss-Prot database (human, release 2022_02) using Mascot Server 2.6.2 (Matrix Science Ltd, UK) were restricted to ±5 ppm (peptide mass tolerance) and used the following parameters: acetyl (protein N-term), carbamidomethyl (C), Gln->pyro Glu (N-term Q), Glu->pyro Glu (N-term E), oxidation (M), propionamide (C), phospho (ST), and phospho (Y) as variable modifications, fragment mass tolerance ±0.8 Da, one missed tryptic cleavage allowed.

Label-free quantification of proteins was carried out as described (*Müller et al., 2016*). In brief, full-scan MS data were processed with MaxQuant v1.6.3 (https://www.maxquant.org) to obtain mass

calibrated peptide signal intensities (peak volumes [PVs]). Their elution times in the evaluated datasets were pairwise aligned and then assigned to peptides based on their m/z and elution time (obtained either directly from MS/MS-based identification or indirectly from identifications in parallel datasets) using in-house developed software. Matching tolerances were ±2 ppm and ±1 min. Abundance$_{norm-spec}$ values reflecting the molecular abundance of proteins were calculated as the sums of all assigned and protein isoform-specific PVs divided by the number of MS-accessible protein isoform-specific amino acids (*Bildl et al., 2012*).

## 2D antibody shift assay

Blue native gel-based antibody shift analysis was carried out as described in *Schmidt et al., 2017*. Two mg of human erythrocyte membrane were solubilised and centrifugated as described for mass spectrometry analysis. 600 µl of the solubilisate was mixed with anti-basigin antibodies (8.4 µg, TA501164 [9H5]+TA501189 [10E10], both Origene) for 2 hr and 600 µl remained untreated. Samples were concentrated on a sucrose cushion (200 µl 20% sucrose in ComplexioLyte 47+protease inhibitors and 300 µl 50% sucrose in 750 mM aminocaproic acid and 50 mM BisTris, pH 7.0) by ultracentrifugation at 400,000 × *g* for 1.5 hr. 300 µl of both sucrose phases, together with 120 µl of solubilised rat brain mitochondria (serving as size marker), were separated overnight on a 2–12% blue native PAGE gradient gel. The lanes were excised, equilibrated in 2× Laemmli buffer with 8 M urea and resolved on second dimension 10% SDS-PAGE gels that were blotted on PVDF membranes. The latter were cut, blocked with 3% skim milk powder in PBS/0.05% Tween-20 and stained with anti-PMCA1/4 (Alomone Labs ACP-005), anti-basigin (Proteintech 11989-1AP), or anti-MCT1 (Santa Cruz Biotechnology SC-365501) followed by HRP-conjugated secondary IgG antibodies specific for respective host species (Santa Cruz Biotechnology) and ECL Prime captured by film. Total protein stains of these western blot membranes (SYPRO Ruby Protein Blot Stain, Bio-Rad) were used for proper alignment of blot signals and visualisation of size marker complexes.

## Broad-range blue native gel analysis of native and reconstituted basigin-PMCA complexes

Blue native gel electrophoresis covering an extended size range (linear acrylamide gradient 1–16%) was carried out as described above resolving the following samples: (i) 120 µl of solubilised rat brain mitochondria (serving as size marker), (ii) 200 µg human erythrocyte membrane solubilised in 200 µl ComplexioLyte 47+protease inhibitors, ultracentrifuged (without sucrose cushion), (iii) solubilised tsA basi/nptn -/- cells (*Schmidt et al., 2017*) transfected with N-terminally Flag-tagged human PMCA4+C-terminally His-tagged human basigin (9 ml ComplexioLyte 47+protease inhibitors/g cells), and (iv) two-step affinity-purified PMCA4-basigin complex (40 µg). The latter was obtained by adding 20 mM imidazole and 1 mM Mg$^{2+}$ to the solubilisate which was then incubated with Ni-Sepharose beads (high performance 17-5268-01, Cytiva) for 90 min. After washing of the Sepharose resin on a column for 30 min (150 mM NaCl, 20 mM Tris-HCl, 30 mM imidazole, 0.05% GDN+protease inhibitors), bound proteins were eluted with three bed volumes of the same buffer adjusted to 200 mM imidazole. The eluate was then incubated with anti-Flag M2 sepharose beads (A2220, Sigma-Aldrich) for 2 hr. After washing on a column for 60 min, proteins were eluted (150 mM NaCl, 20 mM Tris-HCl, 0.05% GDN+triple Flag peptide [0.025%]). The eluate was then concentrated to 2 mg/ml by ultrafiltration (Vivaspin 6/100,000 MW, Sartorius) and ultracentrifuged for 15 min at 150,000 × *g* to remove protein aggregates. Lanes (ii–iv) were excised and subjected to SDS-PAGE/western blot analysis as described above except that a chemiluminescence reader (ImageQuant 800, Amersham) was used for imaging.

## Expression and purification of BAP-tagged PfRH5

A construct was generated consisting of PfRH5 with an N-terminal BAP and a C-terminal C-tag, inserted into the pExpreS2.1 vector (ExpreS2ion Biotechnologies, Hørsholm, Denmark). This was transfected into Schneider 2 (S2) cells and after 24 hr zeocin was added to a final concentration of 0.05 mg/ml. A polyclonal cell line was selected over 3 weeks and was then expanded to higher volumes for isolation and purification.

1.5 l of cell supernatant were concentrated 10-fold, and buffer exchanged to Tris-buffered saline (TBS), pH 7.8 using four omega T-series centramate 0.1 sq. ft tangential filter flow unit (3 kDa, PALL). The buffer-exchanged supernatant was loaded onto a CaptureSelect C-tagXL pre-packed column

(Thermo Fisher Scientific). The column was washed with 50 ml of TBS, pH 7.8 and PfRH5 was eluted with buffer containing 20 mM Tris-Cl, pH 7.8 and 2 mM $MgCl_2$ as six fractions of 500 µl each. Each fraction was quantified at 280 nm using a nano-spectrophotometer (Thermo Fisher Scientific) and fractions of concentration 0.5 mg/ml or more were subjected to separation on a Superdex 200 increase 10/300 GL column (cytiva) for further purification.

## Purification of the PfRH5-PfCyRPA- PfRIPR (RCR) complex

PfRH5, PfCyRPA, and PfRIPR were expressed and purified as described earlier (*Ragotte et al., 2022*). After purification PfRH5, PfCyRPA, and PfRIPR were mixed in an equimolar ratio and incubated on ice for 30 min. The assembled complex was then injected into a Superdex 200 increase 10/300 GL column (cytiva) to obtain a homogenous, purified PfRCR complex.

## Purification of basigin-PMCA complex from ghosts

100 ml of frozen human ghost cells were thawed in a water bath at 20°C. Ghosts were then solubilised by adding DDM/CHS solution to final concentration of 1.2% at 4°C under gentle stirring for 1 hr. Non-solubilised material was removed by centrifugation at 100,000 × *g* for 30 min and $CaCl_2$ was added to the supernatant to final concentration of 0.25 mM. One ml of CaM resin (MERCK) pre-equilibrated in wash buffer containing 20 mM HEPES-Na, pH 7.2, 300 mM NaCl, 10% glycerol, 0.25 mM $CaCl_2$, and 1.2% DDM/CHS was added to the supernatant mixture and incubated at 4°C under gentle stirring. After 1 hr, the resin was transferred to a gravity flow column and washed with 50 ml of equilibration buffer containing 0.02% instead of 1.2% DDM/CHS. Finally, protein bound to CaM resin was eluted in 8–10 fractions of 250–300 µl each in total 2.4 ml of elution buffer containing 20 mM HEPES-Na, pH 7.2, 150 mM NaCl, 1 mM EDTA, and 0.02% DDM/CHS. Identity and purity of eluted proteins was assessed by coomassie staining on 10% SDS-PAGE followed by western blotting with mouse anti-human basigin IgG (Biolegend). Fractions were pooled and concentrated using Amicon centrifugal filter units 100 kDa (Merck) to separate on a Superdex 200 increase 10/300 GL column equilibrated (GE Healthcare) with buffer containing 20 mM HEPES-Na, pH 7.2, 150 mM NaCl, and 0.02% DDM/CHS to check homogeneity or integrity of affinity-purified complex. Using this procedure, a total of 20–35 µg of basigin-PMCA complex was obtained.

## Recombinant expression and purification of basigin-MCT1 complex

Synthetic genes (GeneArt) encoding full-length human MCT1 (uniprot ID: P53985) and human basigin (uniprot ID: P35613-2) were cloned individually into pFastBac vector (Invitrogen) for recombinant expression in Sf9 cells. MCT1 was fused to a C-terminal His$_6$-tag, whereas basigin was not tagged. Bacmids were generated in DH10Bac cells (Life Technologies) and purified by isopropanol precipitation using a miniprep kit (QIAGEN). First-generation baculoviruses (P1) were produced by transfecting Sf9 cells with individual bacmids at a cell density of 1×10$^6$ cells/ml and subsequently amplified to produce third-generation viruses (P3). Expression of the complex was induced by adding an equal number of viral particles from both MCT1 and basigin-containing P3 viruses to Sf9 cells (~2.5–3.0 × 10$^6$ cells/ml). After 48 hr cells were harvested and resuspended in lysis buffer (25 mM Tris, pH 8.0, 150 mM NaCl, 10% glycerol, and EDTA-free complete protease inhibitor) and lysed using a Dounce homogeniser sonication (60% amplitude, 3 s 'on' and 9 s 'off' pulse for 1:30 min) on ice. Lysed homogenate was centrifuged at 3000 × *g* for 20 min to remove cell debris, and the obtained supernatant was spun at 100,000 × *g* for 45 min at 4°C to isolate Sf9 plasma membranes. Isolated membranes homogenised in lysis buffer were solubilised with 1.2% DDM/CHS at 4°C for 1 hr. After centrifugation at 100,000 × *g* for 45 min, the supernatant was incubated with Ni-NTA resin (QIAGEN) at 4°C for 1 hr. The resin was then passed through on a gravity flow column, which was washed with 25 mM Tris, pH 8.0, 300 mM NaCl, 10% glycerol, 15 mM imidazole, and 0.02% DDM/CHS. The protein was eluted in lysis buffer supplemented with 400 mM imidazole. Eluted protein complex was further purified on a Superdex 200 increase 10/300 GL column, and a total of 0.2–0.3 mg could be obtained from 500 ml Sf9 cell culture.

## Recombinant expression and purification of full-length basigin from Sf9 cells

A synthetic gene encoding human basigin (uniprot ID: P35613-2) was cloned into pFastBac vector for recombinant expression as C-terminal His$_6$-tagged membrane protein in Sf9 cells (Sigma). Bacmid containing the target gene was generated in DH10Bac cells (Life Technologies) and purified by isopropanol precipitation using lysis and neutralisation buffers from a plasmid miniprep kit (QIAGEN). First-generation baculoviruses (P1) were produced by transfecting Sf9 cells with individual bacmids at a cell density of $1\times10^6$ cells/ml and subsequently amplified to produce second-generation viruses (P2). Full-length basigin was expressed by adding P2 viruses to Sf9 cells (~2.5–3.0 × $10^6$ cells/ml) at a final concentration of 1%. After 48 hr, cells were harvested and resuspended in lysis buffer (25 mM Tris, pH 8.0, 150 mM NaCl, 10% glycerol, and EDTA-free complete protease inhibitor) and lysed using a Dounce homogeniser followed by sonication (60% amplitude, 3 s 'on' and 9 s 'off' pulse for 1:30 min) on ice. The lysed homogenate was centrifuged at 3000 × $g$ for 20 min to remove cell debris, and the obtained supernatant was spun at 100,000 × $g$ for 45 min at 4°C to isolate Sf9 plasma membranes. Isolated membranes corresponding to 500 ml of Sf9 cell culture were first resuspended in 50 ml lysis buffer using a Dounce homogeniser and were then solubilised with 1.2% (vol/vol) DDM:CHS at 4°C for 1 hr. After centrifugation at 100,000 × $g$ for 45 min, the supernatant was incubated with Ni-NTA resin (QIAGEN) at 4°C for 1 hr. The resin was then passed through on a gravity flow column, which was washed with 25 mM Tris, pH 8.0, 300 mM NaCl, 10% glycerol, 15 mM imidazole, and 0.02% DDM:CHS. The protein was eluted in lysis buffer supplemented with 400 mM imidazole. Eluted protein was further purified on an S200 10/300 column (Cytiva), and a total of 0.2–0.4 mg could be obtained from 100 ml Sf9 cell culture.

## Purification and production of full-length basigin in monomeric form

Isolated membranes corresponding to 250 ml of Sf9 cells containing His$_6$-tagged full-length basigin were resuspended using Dounce homogeniser in 50 ml of buffer containing 20 mM Tris-HCl, pH 8.0, and 150 mM NaCl. Membranes were then solubilised by dropwise adding 10% (vol/vol) CHAPS to a final concentration of 1–1.2%. Solubilisate was cleared by centrifuging at 100,000× $g$ for 45 min, and the supernatant was incubated with Ni-NTA resin (QIAGEN) at 4°C for 1 hr. The resin was then passed through a gravity flow column, which was washed with 25 mM Tris-HCl, pH 8.0, 300 mM NaCl, 15 mM imidazole, and 1% CHAPS. The protein was eluted in lysis buffer supplemented with 400 mM imidazole. Eluted protein was purified on a superdex 200 10/300 increase column (Cytiva) resulting in 0.1–0.2 mg of monomeric full-length basigin.

To incorporate CHAPS-purified monomeric full-length basigin into DDM:CHS, purified protein was first immobilised on Ni-NTA by incubating with the resin for 2 hr in a cold room. The resin was then washed at 0.5 ml/min flow rate with 50–60 ml of buffer containing 20 mM Tris, pH 8.0, 150 mM NaCl, and 0.1% DDM:CHS. After washing, the protein was eluted and applied to a superdex 200 10/300 increase column (Cytiva), pre-equilibrated with 20 mM Tris-HCl, pH 8, 150 mM NaCl, and 0.02% DDM:CHS. Peak fractions corresponding to non-aggregated, monomeric protein were pooled to produce 0.03–0.05 mg of micelle-exchanged protein.

## SPR affinity measurements

SEC-purified PfRH5 was biotinylated enzymatically using *Escherichia coli* biotin ligase to transfer a single biotin residue. Experiments were performed at 25°C on a Biacore T200 instrument (GE Healthcare) using a CAP chip (Biotin CAPture kit [Cytiva]) in a buffer containing 20 mM HEPES-Na, pH 7.5, 150 mM NaCl, 1 mM EDTA, 0.02% DDM/CHS, and 1 mg/ml salmon sperm DNA. Purified analyte proteins were equilibrated in the SPR buffer using a PD-5 column prior to the experiment. The sensor surface was first coated with oligonucleotide-coupled streptavidin following the manufacturer's instructions. 50–70 RUs of monobiotinylated PfRH5 were captured on the chip to quantify interactions with basigin-PMCA and basigin-MCT1 complexes, whereas 300–350 RUs were immobilised for basigin ectodomain binding. Binding measurements were performed at 30 µl/min by injecting twofold dilution series from 400 nM to 1.56 nM for basigin-PMCA, 1.6 µM to 6.25 nM for basigin-MCT1 complexes, and 4 µM to 7.81 nM for basigin ectodomain. Association was measured for 180 s followed by dissociation for 300 s and after each binding cycle, the sensor chip surface was regenerated by injecting 10 µl of 6 M guanidium-HCl and 1 M NaOH, pH 11.0 mixed in a 4:1 ratio. The data were

processed using BIA evaluation software version 1.0 (BIAcore, GE Healthcare) and response curves were double referenced by subtracting the signal from both reference cell and averaged blank injections. Kinetic constants were calculated using global analysis, fitting 1:1 Langmuir model (A+B ↔AB) for basigin ectodomain and full-length basigin, and a two-state model involving conformation change (A+B ↔AB* ↔ AB) for basigin-PMCA and basigin-MCT1 interactions with PfRH5.

### SPR analysis using ghost proteins

SEC-purified PfRH5 in 1× PBS was chemically biotinylated using EZ-Link Sulfo-NHS-Biotin (Thermo Fisher Scientific) following the manufacturer's instructions. Basigin-enriched ghost protein and its PMCA and MCT1 depleted versions were first exchanged into SPR buffer on a PD-5 column. Each ghost fraction was then flowed individually over CAP chip surface coated with 400–450 RUs of chemically biotinylated PfRH5. The association phase was measured for 180 s, while the dissociation phase was observed for 300 s and the recorded response curves were double referenced by subtracting response data from reference surface and buffer blank.

### SPR analysis of monoclonal antibody blocking

450–500 RUs of chemically biotinylated PfRH5 were immobilised on flow cell 2 of the CAP chip. A mAb blocking experiment was then performed by injecting a single concentration (typically 50 nM for each mAb) for 240 s followed immediately by an analyte injection (2 µM basigin ectodomain or 0.2–0.4 µM basigin-PMCA or basigin-MCT1 complex) for 180 s on flow cells 1 and 2. Also, 10-fold dilution series of 10 µM to 0.1 nM of R5.016 was applied for 240 s to gauge concentration-dependent inhibition of the binding of PfRH5 to basigin-PMCA by R5.016. All sensogram curves were double referenced by subtracting responses from reference surface and blank injections.

### Measurement of Ca$^{2+}$-ATPase activity

0.075–0.1 µg of basigin-PMCA was incubated in a buffer containing 20 mM HEPES-NaOH, pH 7.2, 150 mM NaCl, 1 mM MgCl$_2$, 0.5 mM EGTA and 0.1 mM ATP and 0.02% (wt/wt) DDM/CHS. The reaction started by addition of 0.55 mM CaCl$_2$, incubated for up to 30 min at 30°C. The reaction was stopped by addition of 1 mM EGTA following which the inorganic phosphate (P$_i$) released due to ATP hydrolysis was detected by using P$_i$ color lock gold kit (Abcam) and by measuring absorbance at 625 nm using a Spectra Max (Molecular Devices) colorimeter. The Ca$^{2+}$-ATPase activity in the reaction mixture was calculated by subtracting the amount of P$_i$ liberated in the tubes without Ca$^{2+}$ and expressed in micromolar at different time points. The concentration of P$_i$ in the reaction mixes was determined from a linear standard curve generated from Pi standard solution provided in the kit.

To study inhibition of Ca$^{2+}$-ATPase activity, 0.1 µg of basigin-PMCA complex in 100 µl reaction buffer was pre-incubated with 0–32 µM of PfRH5 or caloxin 1c2 peptide in a 96-well microplate. The ATPase activity was stimulated by addition of 0.55 mM CaCl$_2$ at 30°C. After 10 min, the reaction was stopped with 1 mM EGTA and colour was developed using Pi color lock gold kit followed by absorbance measurements at 625 nm. The inhibition of PMCA activity was inferred by comparing amount of P$_i$ liberated in the reaction mixes containing PfRH5 or caloxin 1c2 to the reaction mix without any inhibitor.

### Electrophysiology

Chinese hamster ovary cells were transiently transfected with cDNA coding for the mouse BK$_{(Ca)}$ channel alpha subunit (Uniprot ID Q08460), PMCA4b (P23634), and Basigin (P35613). The cells were incubated at 37°C and 5% CO$_2$ and measured 2–4 days after transfection. Whole-cell patch clamp recordings were performed at room temperature using a HEKA EPC 10 amplifier. The currents were low pass filtered at 3–10 kHz and sampled at 20 KHz. Leak currents were subtracted using a P/4 leak subtraction protocol with holding potential of –90 mV and voltage steps opposite in polarity to those in experimental protocol. Serial resistance was 30–70% compensated using the internal compensation circuitry. The standard extracellular solution contained 5.8 mM KCl, 144 mM NaCl, 0.9 mM MgCl$_2$, 1.3 mM CaCl$_2$, 0.7 mM NaH$_2$PO$_4$, 5.6 mM D-glucose, and 10 mM HEPES-NaOH, pH 7.4. Recording pipettes pulled from quartz glass had resistance of 2.5–4.5 MΩ when filled with internal solution containing 139 mM KCl, 3.5 mM MgCl$_2$, 2 mM DiBrBAPTA, 5 mM HEPES-NaOH, pH 7.3; 2.5 mM Na$_2$ATP, 0.1 mM Na$_3$GTP, pH 7.3. CaCl$_2$ was added to the internal solution to obtain 5 µM free [Ca$^{2+}$].

PfRH5 and PfRCR were applied at final concentration of 2 and 1 µM respectively, for 3–10 min in the bath before recording.

Steady-state activation of $BK_{Ca}$ channels was determined using test pulses ranging from –50 to +200 mV (in 10 mV increment), followed by a repolarisation step to 0 mV. The conductance-voltage relations have been determined from the tail current amplitudes measured 0.5 ms after repolarisation to the fixed membrane potential (0 mV) and normalised to the maximum. For fitting, a Boltzmann function was used: $g/g_{max} = g_{max}/(1+\exp((V_h-V_m)/k))$, where $V_h$ is the voltage required for half maximal activation and k the slope factor.

All the chemicals except DiBrBAPTA (Alfa Aesar) were purchased from Sigma.

## Immunofluorescence

100 µl of packed erythrocytes were washed by resuspending cells in 400 µl of 1× PBS and pelleted at 1500 × $g$ for 5 min at 4°C. After three washes, cells were fixed using a 4% paraformaldehyde solution (Thermo Fisher Scientific) containing 0.0075% glutaraldehyde (Merck) for 45 min at room temperature. Cells were washed three times using 1× PBS and treated with 0.1 mg/ml sodium boro-hydride for 30 min at room temperature (Merck). Treated cells were washed three times with 1× PBS and were blocked afterwards with 4% BSA solution for 1 hr at 4°C. 20 µl erythrocytes were incubated with anti-basigin (1:200), anti- PMCA (1:50), and anti-MCT1 (1:50) antibodies overnight at 4°C. All antibodies tested were acquired from commercial sources: anti-TRA-1–85/CD147 antibody (R&D Systems, MAB3195-SP), anti-MCT1 polyclonal antibody (Almone Labs LTD, distributed by Thermo Fisher Scientific, AMT-011) and anti-PMCA polyclonal antibody (Invitrogen, distributed by Thermo Fisher Scientific, PA5-107073). Antibody-bound cells were washed three times with 1× PBS with 0.05% Tween 20, followed by labelling with anti-mouse IgG Alexa 488 (1:300) and anti-rabbit IgG (1:300). Labelled cells were washed another three times with 1× PBS with 0.05% Tween 20 and resuspended in 300 µl of 1× PBS followed by incubation on a coverslip coated with 0.1 mg/ml poly-D lysine 30 min. Coverslips containing mounted erythrocytes were placed on a glass slide and sealed using nail paint. Imaging was performed using an Oxford Nano imager microscope using a 100× oil immersion objective equipped with continuous wave excitation laser 483 nm and 540 nm. Images were obtained at an exposure time of 200 ms with an objective-based total internal reflection angle of 54 degrees, using Nano imager software version 1.7, and the images were analysed using ImageJ.

## *P. falciparum* culture and growth inhibition assay

*P. falciparum* strain 3D7 was cultured using human erythrocytes (supplied by NHSBT at the John Radcliffe Hospital, Oxford, UK) at 2% hematocrit in RPMI 1640 (Thermo Fisher Scientific, 21870076) supplemented with 25 mM HEPES, 50 mg/l hypoxanthine, 2 g/l D-glucose, 2 mM L-glutamine, 10 µg/ml gentamycin, and 0.5% (wt/vol) AlbuMAX II (Thermo Fisher, 11021029) at 37°C in static cell culture flasks gassed with 5% $CO_2$ and 5% $O_2$. For growth inhibition assays, late-stage schizont cultures, synchronised by serial treatment with 5% sorbitol, were incubated in 96-well plates for one complete life cycle (ca. 44 hr) with antibodies at various concentrations in 40 µl at 1% hematocrit and an initial parasitaemia of 0.4%. All antibodies tested were acquired from commercial sources: anti-TRA-1–85/CD147 antibody (R&D Systems, MAB3195-SP), anti-MCT1 polyclonal antibody (Almone Labs LTD, distributed by Thermo Fisher Scientific, AMT-011), and anti-PMCA polyclonal antibody (Invitrogen, distributed by Thermo Fisher Scientific, PA5-107073). Parasite growth was assessed by measuring parasite lactate dehydrogenase activity as described previously (*Makler and Hinrichs, 1993*), using control cultures with only medium or medium supplemented with 5 mM EDTA as references for 0% and 100% growth inhibition, respectively.

## Data processing software

ImageJ was used for densitometric analysis of western blots. SPR curves, SEC elution traces, and other graphs were reproduced using Prism 9 software and structural figures were prepared with Pymol (Schroedinger).

## Acknowledgements

This work was funded by a Wellcome Investigator Award (20797/Z/20/Z) to MKH and was supported by grants of the DFG (SFB 746, TP 16, Fa 332/9-1) to BF and (SFB 746, TP 20) to US. The authors would

like to thank David Staunton for support with biophysics data collection and Dr. Jagadish Prasad Hazra (University of Oxford) for help with microscopy data acquisition.

## Additional information

### Competing interests

Sebastian Henrich: is affiliated with Roche Pharma AG. The author has no financial interests to declare. Bernd Fakler: is a shareholder of Logopharm GmbH. Logopharm GmbH produces ComplexioLyte 47 used in this study. The company provides ComplexioLyte reagents to academic institutions on a non-profit basis. Simon J Draper: is a named inventor on patents related to PfRH5-targeting antibodies.(PCT/GB2105/052205, PCT/GB2017/052608 and PCT/GB2019/052885). Uwe Schulte: is an employee and shareholder of Logopharm GmbH and BF is shareholder of Logopharm GmbH. Logopharm GmbH produces ComplexioLyte 47 used in this study. The company provides ComplexioLyte reagents to academic institutions on a non-profit basis. Matthew K Higgins: US is an employee and shareholder of Logopharm GmbH and BF is shareholder of Logopharm GmbH. Logopharm GmbH produces ComplexioLyte 47 used in this study. The company provides ComplexioLyte reagents to academic institutions on a non-profit basis.named inventor on patents related to PfRH5-targeting antibodies (PCT/GB2105/052205, PCT/GB2017/052608 and PCT/GB2019/052885). The other authors declare that no competing interests exist.

### Funding

| Funder | Grant reference number | Author |
| --- | --- | --- |
| Wellcome Trust | 20797/Z/20/Z | Abhishek Jamwal Matthew K Higgins Stephan Hirschi |
| Deutsche Forschungsgemeinschaft | SFB 746 | Bernd Fakler |
| Deutsche Forschungsgemeinschaft | TP 20 | Bernd Fakler |

The funders had no role in study design, data collection and interpretation, or the decision to submit the work for publication. For the purpose of Open Access, the authors have applied a CC BY public copyright license to any Author Accepted Manuscript version arising from this submission.

### Author contributions

Abhishek Jamwal, Conceptualization, Formal analysis, Investigation, Writing – original draft, Writing – review and editing; Cristina F Constantin, Conceptualization, Formal analysis, Investigation, Writing – original draft; Stephan Hirschi, Conceptualization, Formal analysis, Investigation, Methodology; Sebastian Henrich, Wolfgang Bildl, Bernd Fakler, Conceptualization, Formal analysis, Investigation; Simon J Draper, Resources; Uwe Schulte, Conceptualization, Resources, Formal analysis, Supervision, Funding acquisition, Writing – original draft, Writing – review and editing; Matthew K Higgins, Conceptualization, Formal analysis, Supervision, Funding acquisition, Investigation, Visualization, Writing – original draft, Writing – review and editing

### Author ORCIDs

Abhishek Jamwal http://orcid.org/0009-0004-6842-6056
Simon J Draper https://orcid.org/0000-0002-9415-1357
Uwe Schulte https://orcid.org/0000-0003-3557-0591
Matthew K Higgins https://orcid.org/0000-0002-2870-1955

### Decision letter and Author response

Decision letter https://doi.org/10.7554/eLife.83681.sa1
Author response https://doi.org/10.7554/eLife.83681.sa2

# Additional files

## Supplementary files
• Supplementary file 1. Binding constants derived from surface plasmon resonance analysis by fitting sensograms with Langmuir 1:1 model.

• Supplementary file 2. Binding constants derived from surface plasmon resonance analysis by fitting sensograms with two-state interaction model.

• MDAR checklist

## Data availability
Data within graphs (source data) and uncropped gel and blot images are included as Source Data files. Additional data is included in Supplementary files 1 and 2.

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
