## [Editor Report]

This elegantly performed and rigorous study generates new and conceptually important insights into the interaction between an essential malaria parasite invasion ligand (and vaccine candidate) called PfRH5, and its erythrocyte surface integral membrane receptor basigin. The authors provide compelling evidence based on rigorous biochemical assays that erythrocyte basigin is predominantly expressed in a complex with one of two distinct erythrocyte membrane proteins called PMCA and MCT1 and that PfRH5 binds to these complexes better than to isolated basigin. Certain invasion-inhibitory antibodies, that do not prevent binding of PfRH5 to isolated basigin, do in contrast prevent binding to the basigin complexes, explaining the mode of action of these previously enigmatic antibodies and providing valuable data towards the improved design of vaccines based on PfRH5.

---

## [Decision Letter]

**Decision letter after peer review:**

Thank you for submitting your article "Erythrocyte invasion-neutralising antibodies prevent *Plasmodium falciparum* RH5 from binding to basigin-containing membrane protein complexes" for consideration by *eLife*. Your article has been reviewed by 3 peer reviewers, and the evaluation has been overseen by a Reviewing Editor and Dominique Soldati-Favre as the Senior Editor. The following individual involved in the review of your submission has agreed to reveal their identity: Michael J Blackman (Reviewer #2).

All three reviewers appreciated that this is an important study that substantially advances our understanding of malaria parasite interactions with human erythrocytes during invasion. While they found the evidence compelling, they also identified two major weaknesses that should be addressed in a revised version of the manuscript:

Essential revisions:

(1) The reviewers expressed some concerns about the calcium flux experiments. The data show that RH5 binding does not alter the calcium pump activity of PMCA in purified complexes or expressed in CHO cells. Unless they can provide experimental evidence that RH5-PMCA interaction has no effect on calcium flux in the context of erythrocyte invasion, they should tone down their claims and not rule out the role of PMCA inhibition during invasion. They should also confirm that recombinant PMCA4b and basigin interact and form the expected complex in their expression system, and consider performing a BKca activation dose-response experiment after the addition of Ca ionophore with and without a buffered low nanomolar [Ca].

(2) The reviewers were concerned that the SPR results may reflect the use of a full-length basigin versus just the ectodomain. This point needs to be addressed to more firmly establish that the association with other proteins is indeed critical for RH5 binding. Also, the reviewers raised some technical issues regarding the evidence for the absence of free monomeric basigin in the SEC and 2D blue native PAGE experiments.

The reviewers made some additional comments to improve the manuscript, as indicated in the individual reviews.

*Reviewer #1 (Recommendations for the authors):*

(1) While the work is extremely well performed and provides compelling evidence that BSG is found in complexes with PMCA or MCT1 and that BSG-PMCA and BSG-MCT1 bind RH5 with higher affinity than isolated BSG, the story would be complete if the authors could include functional data on the role of PCMA and MCT1 during *P. falciparum* invasion. Have they tested the invasion-neutralising activity of specific antibodies and/or chemical inhibitors (such as caloxin 1c2)?

(2) The authors rule out the possibility that RH5 binding to BSG-PMCA may contribute to the calcium spike observed during erythrocyte invasion, based on the observation that RH5 does not alter the calcium pump activity of PMCA in purified complexes or expressed in CHO cells. However, formal demonstration would require evidence that RH5-PMCA interaction has no effect on calcium flux in the context of erythrocyte invasion. The authors may consider inhibiting PMCA and testing for calcium flux in invasion experiments. In the absence of such experimental data, they should tone down their claim.

(3) The antibody shift assay in figure 1 not only shows that basigin is essentially complexed with PMCA and MCT1, but also that all PMCA and all MCT1 are complexed with basigin (Figure 1b). This is quite surprising and should be discussed. Is it possibly due to partial solubilization by DDM/CHS? Along the same line, how do the authors explain that basigin, PMCA, and MCT1 are all found in the same fraction (7-9), although the expected size of each complex is different (BSG-MCT1 would rather be expected in fractions 11-12)? Have the authors tested other detergents and/or cross-linking to check that BSG is exclusively found in complexes with PMCA or MCT1? Another possible control is to test the effect of anti-MCT1 antibody on PMCA-BSG in mobility shift assays (supplementary Figure 1).

*Reviewer #2 (Recommendations for the authors):*

The study and its conclusions are conceptually simple but rigorous and elegantly performed. The experimental approaches used are appropriate and the data are not over-interpreted. I have no major criticisms of the work. However, I have just a few comments regarding issues that might improve the manuscript.

1. Figure 1a and 1b: for the experiments using SEC or 2D blue native PAGE to analyse the form of basigin in erythrocyte ghost extracts, repeated reference is made to the predicted position of migration of monomeric free basigin. However, the authors do not calibrate their SEC or PAGE systems with this protein so this is a prediction only. I am aware that most of their work on recombinant basigin has used just the protein ectodomain, but do they also have access to recombinant full-length basigin (i.e. including its TMD) for this purpose?

2. The evidence presented that PfRH5 binds with higher affinity (slower off-rates, as well as apparently slower on-rates) to the native basigin-PMCA and basigin-MCT1 complexes than to recombinant basigin itself is convincing. However, this is discussed only briefly and the authors do not really attempt to provide a plausible explanation for this, since there is no structural evidence that RH5 makes molecular contacts with the basigin partner proteins. Is it conceivable that PfRH5 makes interactions with segments of PMCA or MCT1 (either protein components or post-translational modifications) that do not present electron density in the available crystal structures? I do feel that this point is worthy of a little more attention in the Discussion.

*Reviewer #3 (Recommendations for the authors):*

1) Figure 1 shows that most basigin on erythrocytes is associated with either MCT1 or PMCA4b. p. 4. The densitometry plot for basigin in Figure 1c does not appear to show a statistically significant reduction with PMCA depletion as the error bars are large relative to the decrease in mean band density. Would statistical analysis of these data clearly establish depletion of basigin with PMCA?

a. The GPC control in this depletion experiment is useful but does not control for nonspecific depletion of basigin on the resin, possibly due to its substantial glycosylation. It would be useful to include depletion of basigin on resin that captures irrelevant proteins (e.g. GPC and Band 3). This is especially important considering that the experiment aims to quantify basigin depletion on sequential resins, where nonspecific adsorption may be cumulative.

2) Figure 1b. p. 4. "Signals at the appropriate molecular weight for free basigin were not observed in either the presence or absence of the basigin-binding antibody" is not clearly supported by the data in Figure 1b because it is not clear that the first dimension (blue native PAGE) reveals proteins migrating at 40-65 kDa. The lowest MW marker, at 180 kDa, is close to the bottom of the 1st dimension migration. A molecular weight standard at or below this range is needed and should be indicated at the top left of the gel.

3) Figure 2e and Supp. Figure 2c. are important because they could reveal changes in erythrocyte Ca++ content resulting from RH5-basigin-PMCA interaction; a change in Ca++ content might reveal why RH5-basigin interaction is required for invasion. The co-expression of BKca, PMCA4b, and basigin in CHO cells is sophisticated and empowered to address this question. The right shift in BKca activation upon PMCA expression is consistent with functional PMCA Ca pumps that efflux Ca++, lowering intracellular Ca required for BKca activation. Controls are however needed to show that the expressed protein PMCA4b and basigin interact and form the expected complex in this system. A control to show that PfRH5 faithfully interacted with this complex upon its exogenous addition is also required to interpret an absence of change. A similar control is suggested for the PfRCR complex experiment (Supp. Figure 2c).

4) Figure 2e and Supp. Figure 2c. Another concern is whether the reduction in intracellular Ca++ with PMCA4b expression has maximally right-shifted the BKca activation curve. If it has done so, then changes in PMCA4b activity upon RH5 (or RCR) binding might not further shift the BKca activation curve in this system. The authors have properly used the Boltzmann equation to fit these curves in Figure 2f and Supp. Figure 2c (per the Methods section), but do not report V1/2 parameters. By eye, my estimate from the data is approximately 130 mV, which is only modestly to the left of the fully right-shifted value reported in PMID: 18955592 (183 mV). Because those workers used a different system, a control is needed here. A straightforward solution to this problem would be to perform a BKca activation dose-response experiment after the addition of Ca ionophore with and without a buffered low nanomolar [Ca] to show that further right shifts are detectable in the authors' hands.

5) Figure 2d, Supp. Figure 3b: "This model is consisted with an interaction which involves two separate, successive events, such as an initial lower affinity capture event, followed by a second binding event which increases the overall affinity, or a conformational change which follows an initial binding event." – some caution is required here because there are other explanations. For example, it is possible that under the SPR conditions basigin-PMCA dissociates into individual subunits. Then, the complex kinetics could reflect the heterogeneous binding of isolated basigin vs intact complexes. Another possibility is the dose-dependent formation of basigin-PMCA aggregates, which would also complicate the observed kinetics. These caveats do not negate the observed tighter binding of basigin-PMCA than of ectodomain alone, but I recommend a more cautious presentation. Some of these caveats also apply to interpreting SPR kinetic data with basigin-MCT1.

6) More fundamentally, the SPR data compare full-length basigin complexed with PMCA or MCT1 to the basigin ectodomain alone and implicitly infer that the difference reflects the formation of a larger complex of basigin with another protein (PMCA or MCT1). Instead, the slow binding could reflect the use of a full-length basigin vs. just the ectodomain. As the RH5 binding sites are well above the membrane and distant from the partner proteins, this reviewer tends to favor better binding results from the use of a full-length basigin (in DDM-CHS, which may stabilize the protein) when compared to the soluble ectodomain fragment. The association with other proteins is compelling and interesting, but may not be critical for RH5 binding or inhibition by antibodies (Figure 4).

7) The docking experiments in Figure 4d and 4e do, however, suggest a contribution from the associated proteins. Antibody inhibition experiments using full-length basigin in DDM-CHS, such as those performed in Figure 4a-c, would be revealing and address this concern definitively, but I do not consider them essential for this publication. The authors may, however, want to present the interpretation of Figure 4a-c more cautiously with this in mind.

8) p. 5. The start of the 3rd paragraph is confusing. "To measure binding kinetics, we produced PfRH5 carrying an N-terminal biotin acceptor peptide tag. This was captured at low density on a streptavidin-coated SPR chip" is in a second paragraph after a paragraph describing SPR with streptavidin-coated chips. I assume the data in the preceding paragraph also used PfRH5 carrying an N-terminal biotin acceptor peptide tag. If correct, the confusion would be corrected by moving the biotin acceptor tag statement to the preceding paragraph and simply describing that dose-response experiments were used to measure binding kinetics in the second paragraph.

9) The Results section for Figure 2f – unclear what method is being used here as it is only described as "colorimetry" in Results and not explained in the figure legend. I believe this is the same ATPase assay as in Figure Suppl 2b, but this should be clarified.

---

## [Author Response]

Essential revisions:1) The reviewers expressed some concerns about the calcium flux experiments. The data show that RH5 binding does not alter the calcium pump activity of PMCA in purified complexes or expressed in CHO cells. Unless they can provide experimental evidence that RH5-PMCA interaction has no effect on calcium flux in the context of erythrocyte invasion, they should tone down their claims and not rule out the role of PMCA inhibition during invasion. They should also confirm that recombinant PMCA4b and basigin interact and form the expected complex in their expression system, and consider performing a BKca activation dose-response experiment after the addition of Ca ionophore with and without a buffered low nanomolar [Ca].

We have now included the control experiments requested. We show, using the 2D gel system, that the PMCA4-basigin complex does form in the cellular system used for the electrophysiology assays and include the data in Figure 2 —figure supplement 2a. We also present data at different calcium concentrations which shows that we can observe right and left shifts in membrane potential and would therefore be able to show the effect of PfRH5 and/or PfRCR on PMCA4:basigin function, should this effect occur (Figure 2 —figure supplement 2b,c). With these controls included, we are confident in our data. Nevertheless, we have been very balanced in the language used in relation to this claim – for example ‘we therefore see no evidence that’ (line 214) and ‘our data do not support this hypothesis’ (line 341) – which we are confident is appropriate.

2) The reviewers were concerned that the SPR results may reflect the use of a full-length basigin versus just the ectodomain. This point needs to be addressed to more firmly establish that the association with other proteins is indeed critical for RH5 binding. Also, the reviewers raised some technical issues regarding the evidence for the absence of free monomeric basigin in the SEC and 2D blue native PAGE experiments.

We have now produced full-length basigin in insect cells, using appropriate detergent to ensure that it is not in complex with other membrane proteins and have conducted SPR experiments using this sample. We present the binding to PfRH5 in Figure 2c, showing that full-length basigin has a very similar affinity and binding kinetics to basigin ectodomain. We have also repeated the antibody inhibition experiments using this full-length basigin and find very similar outcomes to when the same experiments were conducted using basigin ectodomain (Figure 4a). These experiments support our conclusion that differential binding and antibody inhibition are the result of the assembly of basigin into complexes with other membrane proteins in erythrocytes.

In addition, we have addressed technical concerns about the SEC and 2D gel experiments. In the case of the former, we have produced full-length basigin from insect cells, making sure that this experiment was well-conducted to ensure that the material is pure and is in the correct detergent micelle. This full-length basigin has a different mobility on a SEC column when compared with basigin extracted from erythrocytes (Figure 1a). In addition, we repeated the 2D gel analysis, to ensure that free basigin would be detected and we present this gel in supporting data (Figure 1 —figure supplement 1c). In both cases, the outcomes of these control experiment fully support our original conclusions and are presented in full.

The reviewers made some additional comments to improve the manuscript, as indicated in the individual reviews.Reviewer #1 (Recommendations for the authors):1) While the work is extremely well performed and provides compelling evidence that BSG is found in complexes with PMCA or MCT1 and that BSG-PMCA and BSG-MCT1 bind RH5 with higher affinity than isolated BSG, the story would be complete if the authors could include functional data on the role of PCMA and MCT1 during *P. falciparum* invasion. Have they tested the invasion-neutralising activity of specific antibodies and/or chemical inhibitors (such as caloxin 1c2)?

We have now conducted the experiment described here, in which we assessed the growth inhibitory activity of commercially available monoclonal antibodies targeting the extracellular domains of either MCT1 or PMCA. In each case, we were able to source a single antibody which binds to the extracellular region of the receptor and we show that neither of these antibodies inhibits the growth of *Plasmodium falciparum* in human blood culture. Without structural insight to show where on PMCAs or MCT1 these antibodies bind, it is hard to conclude why these antibodies are not inhibitory. This would require a major study in which we generate panels of antibodies and structurally characterise them as well as determining growth inhibitory activity. As MCT1 and PMCAs are unlikely to be therapeutic targets in malaria, it is not our view that this effort would be justified. For experiments related to PMCA inhibition, see below.

2) The authors rule out the possibility that RH5 binding to BSG-PMCA may contribute to the calcium spike observed during erythrocyte invasion, based on the observation that RH5 does not alter the calcium pump activity of PMCA in purified complexes or expressed in CHO cells. However, formal demonstration would require evidence that RH5-PMCA interaction has no effect on calcium flux in the context of erythrocyte invasion. The authors may consider inhibiting PMCA and testing for calcium flux in invasion experiments. In the absence of such experimental data, they should tone down their claim.

Our view is that the experiment proposed here will be easy to do, but very difficult to interpret. The experiment proposes to inhibit PMCA using a chemical inhibitor, such as caloxin, and then assess the effect on invasion. As PMCA is a calcium extruder, its inhibition will lead to a global calcium increase in the RBC. This is not the same as the localised and transient inhibition of PMCA by PfRH5 just at the RBC-merozoite junction, which may lead to a transient flash of calcium. We have not been able to design an experiment in which we locally inhibit PMCA just at this junction. As the global modulation of PMCA before invasion, rather than transient modulation during invasion, will not test the same hypothesis, our view is that this experiment would be misleading, and we prefer not to conduct this study.

We have therefore made a few adjustments to the text, for example ‘making this unlikely as the mechanism’ in line 9 to tone down the claim. Elsewhere, the claim was already ‘toned down’ – for example ‘We therefore see no evidence’ in line 213-4 and ‘Our data do not support this hypothesis’ in line 341, which seem to us correct and balanced.

3) The antibody shift assay in figure 1 not only shows that basigin is essentially complexed with PMCA and MCT1, but also that all PMCA and all MCT1 are complexed with basigin (Figure 1b). This is quite surprising and should be discussed. Is it possibly due to partial solubilization by DDM/CHS? Along the same line, how do the authors explain that basigin, PMCA, and MCT1 are all found in the same fraction (7-9), although the expected size of each complex is different (BSG-MCT1 would rather be expected in fractions 11-12)? Have the authors tested other detergents and/or cross-linking to check that BSG is exclusively found in complexes with PMCA or MCT1? Another possible control is to test the effect of anti-MCT1 antibody on PMCA-BSG in mobility shift assays (supplementary Figure 1).

The strength of our approach to study of the complexation of basigin with PMCAs and MCT1 comes from the use of multiple methods to ask the same question. We therefore include the size exclusion data which the reviewers highlight (Figure 1a), 2D gel analysis (Figure 1b), successive immuno-depletion studies (Figure 1c) and an antibody shift experiment (Figure 1 —figure supplement 2). These experiments use different approaches and protein solubilised with different detergents. In particular, the 2D gel study used a different solubilisation mixture to the others. While there are subtle differences between the outcomes of these assays, the overall picture is very similar. In some assays, all detectable basigin is bound to PMCA4 or MCT1 (ie Figure 1b) while in others (Figure 1c) we find that the majority is bound, perhaps with a small amount of unbound basigin due to the longer wash steps allowing time for dissociation or the different detergent used. By using four different methods to test the same hypothesis, we can be confident that our findings are rigorous and that basigin is found complexed with either PMCAs or MCT1.

To help us to understand the elution profile of the basigin:MCT1 and basigin:PMCA complexes in Figure 1, we can also compare with the elution profiles for purified basigin:PMCA (Figure 2b) and purified basigin:MCT1 (Figure 3d) on similar SEC columns. This comparison shows that each purified complex predominantly elutes between 10-11ml, which is where Western blotting shows each protein also elutes in Figure 1a. This analysis helps us to be confident that we are studying basigin:MCT1 and basigin:PMCA complexes in Figure 1a, despite the potential for purification to alter the micelle properties or lead to loss of associated lipids.

Reviewer #2 (Recommendations for the authors):The study and its conclusions are conceptually simple but rigorous and elegantly performed. The experimental approaches used are appropriate and the data are not over-interpreted. I have no major criticisms of the work. However, I have just a few comments regarding issues that might improve the manuscript.1. Figure 1a and 1b: for the experiments using SEC or 2D blue native PAGE to analyse the form of basigin in erythrocyte ghost extracts, repeated reference is made to the predicted position of migration of monomeric free basigin. However, the authors do not calibrate their SEC or PAGE systems with this protein so this is a prediction only. I am aware that most of their work on recombinant basigin has used just the protein ectodomain, but do they also have access to recombinant full-length basigin (i.e. including its TMD) for this purpose?

We have now included the calibration of the SEC traces in Figure 1a using full-length basigin which we express in baculovirus-infected insect cells. It was not straight-forward to obtain full-length basigin which was not bound to other membrane proteins, as purification in the mild detergent mixture, DDM:CHS also led to the co-purification of membrane protein complexes from the cells. In contrast, purification with the harsher detergent, CHAPS, disrupted these complexes, but CHAPS has a different micelle size to DDM:CHS, which would affect its mobility on SEC. To obtain the right calibration sample, we therefore purified full-length basigin from insect cells in CHAPS and then detergent exchanged into DDM:CHS, to generate a calibration sample in the same detergent as the erythrocyte purified protein complexes of interest. As we expected, this eluted later from a SEC column than the erythrocyte complexes of interest. This calibration has been added to Figure 1a and the data described here in relation to the purification of full-length basigin is in Figure 1 —figure supplement 1a,b.

2. The evidence presented that PfRH5 binds with higher affinity (slower off-rates, as well as apparently slower on-rates) to the native basigin-PMCA and basigin-MCT1 complexes than to recombinant basigin itself is convincing. However, this is discussed only briefly and the authors do not really attempt to provide a plausible explanation for this, since there is no structural evidence that RH5 makes molecular contacts with the basigin partner proteins. Is it conceivable that PfRH5 makes interactions with segments of PMCA or MCT1 (either protein components or post-translational modifications) that do not present electron density in the available crystal structures? I do feel that this point is worthy of a little more attention in the Discussion.

We agree with the reviewer that the biphasic kinetics which we observe are potentially interesting. Our finding that increased association times correlate with decreased dissociation rates is also supportive of a model in which there is an initial weaker interaction which ‘matures’ into a higher affinity interaction. However, SPR data alone is not sufficient to allow us to propose a molecular mechanism as there are multiple mechanisms which can lead to such a kinetic profile. While we are keen not to overinterpret this finding, we do, in lines 322-332 of the discussion, speculate that the observed binding kinetics might be the result of either a conformational change or a second component to the interaction, but we prefer not to comment more precisely on this.

Reviewer #3 (Recommendations for the authors):1) Figure 1 shows that most basigin on erythrocytes is associated with either MCT1 or PMCA4b. p. 4. The densitometry plot for basigin in Figure 1c does not appear to show a statistically significant reduction with PMCA depletion as the error bars are large relative to the decrease in mean band density. Would statistical analysis of these data clearly establish depletion of basigin with PMCA?

We have now repeated this analysis and show the new data in Figure 1a. The error bars are still quite high and a Mann-Whitney test gives p=0.100 for basigin in the starting material vs PMCA depletion. Together with equivalent outcomes from different assays (i.e. Figure 1 —figure supplement 2) we are confident about our overall conclusions.

a. The GPC control in this depletion experiment is useful but does not control for nonspecific depletion of basigin on the resin, possibly due to its substantial glycosylation. It would be useful to include depletion of basigin on resin that captures irrelevant proteins (e.g. GPC and Band 3). This is especially important considering that the experiment aims to quantify basigin depletion on sequential resins, where nonspecific adsorption may be cumulative.

We have conducted this experiment and it is now shown in Figure 1 —figure supplement 1e. Very little basigin binds to resin not coupled with antibody.

2) Figure 1b. p. 4. "Signals at the appropriate molecular weight for free basigin were not observed in either the presence or absence of the basigin-binding antibody" is not clearly supported by the data in Figure 1b because it is not clear that the first dimension (blue native PAGE) reveals proteins migrating at 40-65 kDa. The lowest MW marker, at 180 kDa, is close to the bottom of the 1st dimension migration. A molecular weight standard at or below this range is needed and should be indicated at the top left of the gel.

We have re-run this experiment and show the resultant 2D gel in Figure 1 —figure supplement 1c, including the region of the denaturing gel in which free basigin would run. This confirms our conclusion that all basigin is found in either MCT1 or PMCA complexes in this assay.

3) Figure 2e and Supp. Figure 2c. are important because they could reveal changes in erythrocyte Ca++ content resulting from RH5-basigin-PMCA interaction; a change in Ca++ content might reveal why RH5-basigin interaction is required for invasion. The co-expression of BKca, PMCA4b, and basigin in CHO cells is sophisticated and empowered to address this question. The right shift in BKca activation upon PMCA expression is consistent with functional PMCA Ca pumps that efflux Ca++, lowering intracellular Ca required for BKca activation. Controls are however needed to show that the expressed protein PMCA4b and basigin interact and form the expected complex in this system. A control to show that PfRH5 faithfully interacted with this complex upon its exogenous addition is also required to interpret an absence of change. A similar control is suggested for the PfRCR complex experiment (Supp. Figure 2c).

We have now confirmed that PMCA4 and basigin express to form a complex in the CHO cell system, using the 2D gel approach. This data is presented in Figure 2 —figure supplement 2a.

4) Figure 2e and Supp. Figure 2c. Another concern is whether the reduction in intracellular Ca++ with PMCA4b expression has maximally right-shifted the BKca activation curve. If it has done so, then changes in PMCA4b activity upon RH5 (or RCR) binding might not further shift the BKca activation curve in this system. The authors have properly used the Boltzmann equation to fit these curves in Figure 2f and Supp. Figure 2c (per the Methods section), but do not report V1/2 parameters. By eye, my estimate from the data is approximately 130 mV, which is only modestly to the left of the fully right-shifted value reported in PMID: 18955592 (183 mV). Because those workers used a different system, a control is needed here. A straightforward solution to this problem would be to perform a BKca activation dose-response experiment after the addition of Ca ionophore with and without a buffered low nanomolar [Ca] to show that further right shifts are detectable in the authors' hands.

Our hypothesis is that addition of PfRH5 might lead to a left-shift, towards a lower membrane potential, rather than a right-shift. As PMCA is a calcium efflux pump, the hypothesis is that the calcium spike which occurs in the erythrocyte on invasion could result from an inhibition of PMCA, with an increase in intracellular calcium due to reduced efflux. In our electrophysiology experiment, the expression of PMCA-basigin has led to a right-shift in membrane potential in BK cells. Should PfRH5 or PfRCR inhibit basigin-PMCA, we would then expect a left-shift in membrane potential back towards the original BK cell line. This would be detected within the dynamic range of the experiment. A right shift would result instead from further activation of PMCA-basigin. To ensure that either shift could be detected within the set-up, we have measured membrane potential at different calcium concentrations (Figure 2 —figure supplement 2b,c) and we observe that 50µM calcium does shift the curve to the right of that observed with PfRH5, showing that, should this activated the PMCA-basigin, this would be detectable in our assay conditions.

5) Figure 2d, Supp. Figure 3b: "This model is consisted with an interaction which involves two separate, successive events, such as an initial lower affinity capture event, followed by a second binding event which increases the overall affinity, or a conformational change which follows an initial binding event." – some caution is required here because there are other explanations. For example, it is possible that under the SPR conditions basigin-PMCA dissociates into individual subunits. Then, the complex kinetics could reflect the heterogeneous binding of isolated basigin vs intact complexes. Another possibility is the dose-dependent formation of basigin-PMCA aggregates, which would also complicate the observed kinetics. These caveats do not negate the observed tighter binding of basigin-PMCA than of ectodomain alone, but I recommend a more cautious presentation. Some of these caveats also apply to interpreting SPR kinetic data with basigin-MCT1.

We appreciate the reviewer’s caution in interpretation of SPR data and in ascribing different complex models and we share this approach. This is why, in the case of the basigin-MCT1 complex, we conducted the experiment in which we allowed binding to PfRH5 for different association times and showed that longer association times leads to shower dissociation rates (Figure 3 —figure supplement 1b). This behaviour is highly consistent with a two-step binding model as longer association times gives time for the second step to take place, leading to the stronger interaction and slower off-rate.

We find that both basigin-PMCA and basigin-MCT1 are stable in solution and we do not observe dissociation of the complexes or aggregation. This does not rule out the possibility that this happens in the SPR flow paths, but we see no reason to think that this is the case based on our experience of working with these proteins. In the experiment with different association rates, aggregation and/or dissociation would need to happen over a time scale of seconds to show the differences that we see. For these reasons we will stick with our current interpretation of this data, albeit describing it in what we are confident is appropriately cautious language (‘i.e. ‘this is compatible with’).

6) More fundamentally, the SPR data compare full-length basigin complexed with PMCA or MCT1 to the basigin ectodomain alone and implicitly infer that the difference reflects the formation of a larger complex of basigin with another protein (PMCA or MCT1). Instead, the slow binding could reflect the use of a full-length basigin vs. just the ectodomain. As the RH5 binding sites are well above the membrane and distant from the partner proteins, this reviewer tends to favor better binding results from the use of a full-length basigin (in DDM-CHS, which may stabilize the protein) when compared to the soluble ectodomain fragment. The association with other proteins is compelling and interesting, but may not be critical for RH5 binding or inhibition by antibodies (Figure 4).

We agree with the reviewer, that a better comparison for our basigin-containing membrane complexes is free full-length basigin, in detergent micelles of a matching composition. As explained in our response to reviewer 1, this was not straight-forward to obtain, as recombinant full-length basigin purified from insect cells in complex with binding partners. However, through use of a harsher detergent, followed by detergent exchange, we were able to obtain full-length basigin in DDM:CHS. We therefore measured the affinity of this material for immobilised PfRH5 and this showed a binding profile and affinity which closely match that of basigin ectodomain. This data is now shown in Figure 2c, replacing that for basigin ectodomain, as it is a better comparator for our basigin-containing membrane protein complexes. We find that the affinity and binding profile for full-length basigin and basigin ectodomain are extremely similar, supporting our original conclusion that differences in affinity and binding kinetics in the PMCA- and MCT1-bound complexes are the result of the presence of the binding partner.

7) The docking experiments in Figure 4d and 4e do, however, suggest a contribution from the associated proteins. Antibody inhibition experiments using full-length basigin in DDM-CHS, such as those performed in Figure 4a-c, would be revealing and address this concern definitively, but I do not consider them essential for this publication. The authors may, however, want to present the interpretation of Figure 4a-c more cautiously with this in mind.

We also agree with the author that full-length basigin is the best comparator for this study and we have now included this data in Figure 4, with the basigin ectodomain data now moved into the supplement. In summary, the R5.016 monoclonal also showed no inhibition of binding of PfRH5 to full-length basigin, while 9AD4 showed a small amount of inhibition, most likely due to minor steric hinderance due to the detergent micelle. In neither case was inhibition observed at the same scale as for basigin-PMCA or basigin-MCT1 complexes, supporting our original conclusions.

8) p. 5. The start of the 3rd paragraph is confusing. "To measure binding kinetics, we produced PfRH5 carrying an N-terminal biotin acceptor peptide tag. This was captured at low density on a streptavidin-coated SPR chip" is in a second paragraph after a paragraph describing SPR with streptavidin-coated chips. I assume the data in the preceding paragraph also used PfRH5 carrying an N-terminal biotin acceptor peptide tag. If correct, the confusion would be corrected by moving the biotin acceptor tag statement to the preceding paragraph and simply describing that dose-response experiments were used to measure binding kinetics in the second paragraph.

We agree with the reviewer that this was not clear. The experiment conducted in paragraph 2 used chemically biotinylated protein, which had been biotinylated on lysine residues. This was then immobilised onto the streptavidin-coated chip. We have clarified this in line 168. This is not the optimal way to conduct the experiment as RH5 molecules will be attached in different orientations to the chip surface. Therefore we added a BAP tag to the N-terminus of PfRH5, allowing specific biotinylation at the terminus. This protein was then used for subsequent experiments, ensuring homogeneous capture of PfRH5 on the surface.

9) The Results section for Figure 2f – unclear what method is being used here as it is only described as "colorimetry" in Results and not explained in the figure legend. I believe this is the same ATPase assay as in Figure Suppl 2b, but this should be clarified.

We have added a clause to lines 209-210 to link the experiment more clearly to the methods and show that we are studying inorganic phosphate release using a colorimetric assay.